# Protein as a Second Language for LLMs

## Abstract

Deciphering the function of unseen protein sequences is a fundamental challenge with broad scientific impact, yet most existing methods depend on task-specific adapters or large-scale supervised fine-tuning. We introduce the "***Protein-as-Second-Language***" framework, which reformulates amino-acid sequences as sentences in a novel symbolic language that large language models can interpret through contextual exemplars. Our approach adaptively constructs sequence–question–answer triples that reveal functional cues without any parameter updates. To support this process we curate a bilingual corpus of 79,926 protein–QA instances spanning attribute prediction, descriptive understanding, and extended reasoning. Empirically, our method delivers consistent gains across diverse open-source LLMs and GPT-4o, achieving up to 15% ROUGE-L improvement (average +6.14%) and even surpassing fine-tuned protein-specific language models. These results highlight that generic LLMs, when guided with protein-as-language cues, can outperform domain-specialized models, offering a scalable pathway for protein understanding in foundation models.

## 1 Introduction

Proteins are indispensable molecular machines of life, driving key functions such as maintaining cell structure and enabling cell communication. Their three-dimensional architectures, catalytic activities, interaction networks, and evolutionary trajectories are all encoded within a linear sequence composed of twenty amino-acid characters (26, 72). Therefore, the core of understanding protein function lies in accurately "reading" and "translating" the biological meaning contained within these amino-acid sequences (11, 27). However, this task is fraught with challenges. Although the amino acid sequence is formally like a language—possessing a fixed character set (over 20 genetically encoded amino acids) and potential grammatical rules (physicochemical laws)—the mapping relationship from the one-dimensional sequence to the three-dimensional structure and function is extremely complex and highly context-dependent (49, 65). Consequently, the central challenge of "what cellular function does an unknown amino acid sequence encode?" still lacks a comprehensive solution.

To address this challenge, research efforts on protein understanding can be broadly categorized into two dominant paradigms: *protein representation learning* and *protein–language alignment modeling*. Protein representation learning sees amino-acid sequences as a standalone modality like language and visual, acquires universal protein representations through self-supervised pre-training on large-scale amino-acid sequences, and then attaches lightweight decoders to predict structure or function (77, 5, 30, 54, 9, 68, 67). While this paradigm excels in the universality of its embeddings and in mining deep sequential patterns, these embeddings still rely on additional "interpreters", *i.e.*, post-processing adapters, to be converted into human-understandable explanations. Protein–language alignment modeling, in contrast, co-trains on paired protein sequences and their textual descriptions, establishing a bidirectional mapping within a shared latent space that enables end-to-end text-based question answering (73, 45, 18, 1, 63, 71). Although this route bypasses downstream adapters, it is intrinsically bound to large-scale paired data and often requires re-fine-tuning whenever the output format or downstream objective shifts. In summary, both of these approaches face bottlenecks of large training data requirements, high computational costs, and limited generalization ability.

**Protein as Second Language.** Reflecting on the human cognitive process, we observe that humans exhibit remarkable efficiency and generalization ability when learning a brand-new symbolic

system (i.e., a new language). The key lies in their ability to rely on and transfer their existing native language knowledge system (16, 23). Given the aforementioned "linguistic" properties of protein sequences—possessing a compositional structure and contextual semantics—and our goal of understanding their function using natural language, we propose a novel perspective: to treat protein sequences as a symbolic system that can be learned and interpreted by large language models (LLMs) as a "second language".

Analogous to how humans acquire a second language, *i.e.*, by encountering new words in context and inferring their meaning and usage, we propose a protein language learning framework in which an LLM acquires protein semantics and reasoning ability through context-driven exposure that grounds sequence patterns in functional and structural examples. This framework adaptively constructs learning contexts for a given protein understanding goal, enabling rapid acquisition of target protein knowledge without additional training or sacrificing generalization. To support effective learning, we constructed a "bilingual" dataset of 79,926 protein-sequence–question–answer triples covering functional, descriptive, and extended-information queries. Across Protein2Text (75), Mol-Instructions(14) and ProtDescribe-QA (22), our framework raises the average ROUGE-L by 6.14% across diverse open-source models and GPT-4o, with a maximum gain of 15%, without any task-specific fine-tuning. Our contributions are as follows:

- We introduce the "***Protein-as-Second-Language***" conceptual framework, which recasts amino-acid sequences as a second language that can be acquired via in-context learning, enabling efficient and generalized protein understanding.

- We construct ***a protein-natural language bilingual dataset*** that spans four task families: attribute-based QA, True or False QA, descriptive-text QA, and extended-information QA, to support effective protein language learning and benchmarking.

- We present a protein language learning framework that adaptively constructs learning contexts for protein understanding, yielding significant gains for both open-source models and GPT-4o, enabling them to outperform domain-specialized models without additional training.

## 2 RELATED WORK

### 2.1 LANGUAGE MODELS IN PROTEIN

Protein representation learning with protein language models (PLMs) extends the Transformer to amino-acid strings, producing dense embeddings for property prediction (19, 5, 13, 20, 7, 9, 10) or generative design (38, 41, 34, 15). Because these models are trained exclusively on amino acid sequences, their outputs remain latent vectors that external classifiers must translate into human-readable function. To obviate this indirection, protein–language alignment modeling has emerged, which jointly connects sequences with textual descriptions via (i) contrastive objectives mapping proteins and sentences into a shared space (74, 67), (ii) bioknowledge-augmented pre-training on curated protein–text corpora (15, 57, 34, 44, 79, 31), or (iii) multi-modal LLMs that graft protein encoders onto frozen language backbones (32, 1, 63, 10, 37, 70). While effective, these approaches entail costly retraining or gradient updates and risk catastrophic forgetting when scaled to larger LLMs (25, 66), prompting a shift toward parameter-efficient adaptation.

### 2.2 PROTEIN QA DATESETS

Datasets that couple proteins with natural-language annotations have become the empirical bedrock for developing protein–text hybrid systems. At present, two complementary families of corpora dominate the landscape. The first centers on protein captioning: given an amino-acid sequence alone, the objective is to generate a concise textual description. Representative instances include the richly annotated Swiss-Prot collection (4), the ProteinKG resource (77) and ProtDescribe (75). The second family targets protein question answering: here, both a sequence and a natural-language query are supplied, and the model is required to synthesize an answer grounded in the provided protein. Curated examples span Mol-Instructions (14), UniProtQA (33), ProteinLMBench (52), VenusX (56) and Protein2Text-QA (22).

## 2.3 IN-CONTEXT PROTEIN LEARNING

In-context learning provides a training-free paradigm for cross-modal reasoning (39), mirroring the exemplar-based inference long used in protein science, where sequence–sequence (3, 6, 53, 62) and multiple-sequence alignments (48, 24) derive function from homology. Building on this exemplar-driven paradigm, recent protein–LLM methods such as ProtEx (51) condition models on biologically similar proteins identified from sequence or embedding space (55, 50, 28, 36, 61, 66). However, these exemplar-selection strategies remain grounded entirely in the protein modality and therefore cannot retrieve exemplars with respect to the content of the natural-language query.

## 3 PROTEIN AS SECOND LANGUAGE

We introduce "Protein-as-Second-Language", a framework that treats amino-acid sequences as a new symbolic system to be learned much like humans acquire a foreign language. Just as learners infer the meaning of unfamiliar words by repeatedly encountering them in context, we construct a *protein–natural language bilingual dataset* (Sec. 3.1) and design an *adaptive context construction mechanism* (Sec. 3.2) to provide such contextual exposure. In this way, our framework enables LLMs to acquire protein semantics through exemplars rather than through extensive re-training.

### 3.1 BILINGUAL DATASET CONSTRUCTION

We curate our bilingual dataset in three steps (Figure 1). Starting from 573,661 Swiss-Prot (4) entries with gene ontology (GO) annotations, we avoid directly converting all annotations, as this would introduce heavy redundancy; instead, we construct a balanced sample. Specifically, (i) we prune the GO-directed acyclic graph (GO-DAG) to obtain representative functional categories and group proteins accordingly (Sec. 3.1.1), (ii) perform bilingual deduplication by clustering sequences within each protein group and sampling proteins with diverse functional annotation (Sec. 3.1.2), and (iii) use DeepSeek-R1 (17) to generate attribute, knowledge, descriptive, and true/false QA pairs, yielding 79,926 high-quality protein–QA triples (Sec. 3.1.3).

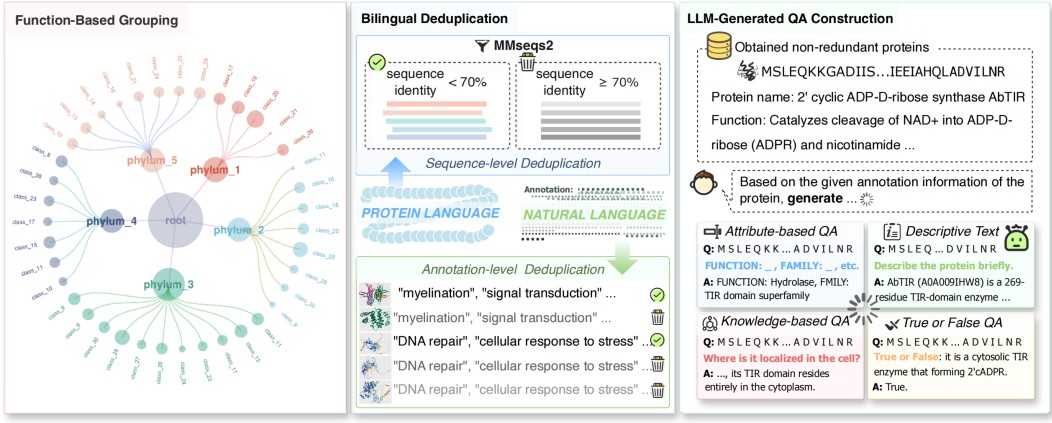

Figure 1: **The overview of data construction of our bilingual protein–QA dataset.**

### 3.1.1 FUNCTION-BASED GROUPING

To enable representative sampling across functional categories, the dataset is partitioned according to the GO hierarchy. Directly using the raw directed acyclic graph (DAG) risks over-fragmentation from overly fine, sparsely populated terms, and excessive generalization near the root. To address this, we adapt a pruning strategy inspired by decision tree simplification (40), where complexity is managed through a penalty to avoid overfitting. This strategy aims to retain an optimal set of GO terms as functional grouping nodes. It balances granularity and coverage, ensuring that the retained nodes represent biologically diverse yet statistically well-supported categories for downstream sampling.

The pruning process is driven by two main criteria: (1) A node is retained if it meets the ***minimum support threshold***, which ensures that the node has a sufficient number of associated proteins, and does not exhibit significant child imbalance. (2) If the ***child-imbalance ratio*** is high, meaning the protein distribution among a node's child terms is uneven, the parent node is retained, even if the child nodes fail to meet the minimum support threshold.

**Minimum Support Threshold**   A node is retained only if the number of associated proteins meets a depth-adjusted threshold $m(d)$, which adapts based on the node's depth in the GO hierarchy. The threshold is calculated as:

$$m(d) = \lambda \cdot C_{tot} \cdot (1 + \beta d) \tag{1}$$

where $C_{tot}$ is the total protein count, $d$ is the node depth, and $\lambda$ and $\beta$ are constants. This dynamic threshold is designed to prevents deep nodes from splitting infinitely due to overly small absolute values.

**Child-Imbalance Ratio**   The child-imbalance ratio is applied to assess whether the child nodes of a given term are too imbalanced. The imbalance ratio $\rho(v)$ is computed as the ratio of the largest to the smallest protein count among the child nodes:

$$p(u) = \frac{\max\limits_{u \in C^+(v)} C(u)}{\min\limits_{u \in C^+(v)} C(u)} \tag{2}$$

where $C^+(v)$ represents the set of valid child nodes with non-zero protein counts. If the imbalance ratio $\rho(v)$ exceeds a specified threshold $\tau(d)$, the parent node $v$ is retained to preserve the biological diversity. This threshold is adjusted dynamically with the depth $d$ to allow for greater flexibility at deeper levels of the hierarchy:

$$\tau(d) = \tau_0 \cdot \alpha^d \tag{3}$$

where $\tau_0$ is the base threshold, and $\alpha$ is a scaling factor.

By applying these two criteria, the pruning process is carried out recursively, allowing the algorithm to adaptively prune the GO DAG and identify the most relevant, biologically diverse functional groups.

### 3.1.2   BILINGUAL DEDUPLICATION

After grouping by GO term, proteins within the same node often exhibit high similarity, as they represent homologous proteins. To address this, we use MMseqs2 (53) for sequence clustering within each GO node, applying a 70% ***amino acid sequence similarity*** threshold. From each cluster, a single representative sequence is selected. This threshold efficiently removes redundant sequences with minimal functional variation while preserving functional diversity.

While sequence similarity-based redundancy removal effectively reduces sequence-level redundancy, it does not necessarily capture functional divergence. Specifically, sequence similarity below 70% does not imply functional divergence, and substantial functional redundancy may still exist within the set (12). To address this, we focus on ***annotation semantic similarity***, quantifying the functional relationships between proteins based on their GO annotations. Inspired by the simGIC method (46) for calculating GO terms semantic similarity, we calculate the Protein Functional Information Content $\text{IC}_{\text{protein function}}$ for each protein, which is the sum of the Information Content (IC) of all associated GO terms and their ancestral terms. The IC of each GO term is calculated based on its frequency in the dataset, using the total protein set after sequence redundancy removal. The $\text{IC}_{\text{protein function}}$ value for each Protein ID is computed as:

$$\text{IC}_{\text{protein function}} = \sum_{g \in \text{GO terms of } p} \text{IC}(g) + \sum_{g' \in \text{ancestors of GO terms of } p} \text{IC}(g'). \tag{4}$$

This provides a quantitative measure of each protein's functional information, capturing both direct and indirect annotations. For each GO term, proteins are sampled based on their unique

IC$_{\text{protein function}}$ values (rounded to 3 decimal places). To ensure balanced species representation, a species quota strategy is applied based on the proportions of Eukaryota, Bacteria, Archaea, and Viruses in the dataset after sequence redundancy removal. This ensures an unbiased species distribution in the final sample. The bilingual deduplication process reduces redundancy in two aspects, amino acid sequence and annotation semantics, ensuring a balanced and diverse protein corpus.

### 3.1.3 LLM-BASED QA CONSTRUCTION

To transform curated protein annotations into natural-language question–answer pairs, we prompt the DeepSeek-R1 (17) model to generate biologically grounded QA texts that reflect both functional attributes and contextual knowledge (the prompts used for each QA type are provided in Appendix E). The resulting QA corpus covers four complementary types: ❶ *Attribute-based QA* captures factual properties directly associated with a protein, such as molecular function, cellular component, or family. ❷ *Knowledge-based QA* comprises concise, annotation-driven questions and answers that involve in multiple biological aspects of a protein, such as expression, localization, mechanism, and interactions. ❸ *Descriptive Text QA* produces longer natural-language explanations that integrate multiple annotations into coherent functional summaries. ❹ *True or False QA* consists of single statements that integrate multiple biological aspects of a protein, accompanied by a True/False answer and a brief explanation.

These four types yield a rich and varied bilingual dataset, ensuring that models are exposed to both concise factual knowledge and more detailed contextual explanations, supporting their ability to understand and reason about protein functions.

## 3.2 BILINGUAL CONTEXTUAL LEARNING

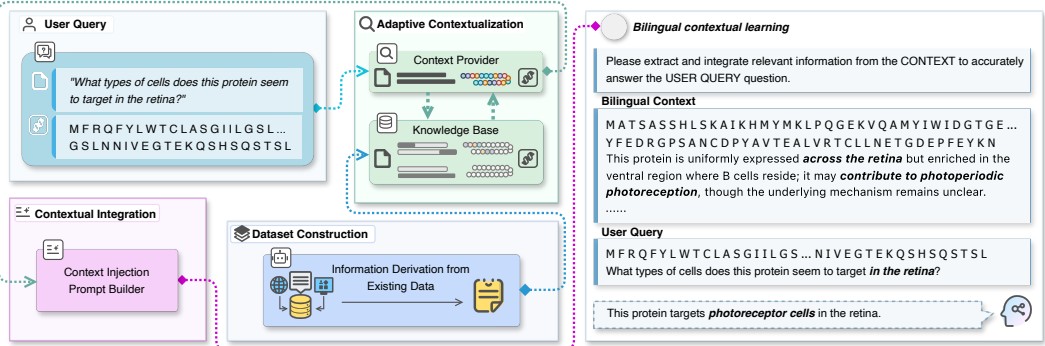

Figure 2: **Process of Query-Adaptive Context Construction.**

In practical scenarios, questions concerning protein sequences are often highly flexible and complex: they require not only analogous proteins with similar sequence patterns to capture potential structural or functional signals, but also complementary descriptive knowledge and QA pairs to provide semantic grounding. As shown in Figure 2, we propose an adaptive context construction mechanism, for ***bilingual contextual learning***, designed to selectively build bilingual learning contexts for each query. Instead of brute-force mixing of amino acid sequences and descriptive texts, the mechanism follows the principle of second language acquisition—exposing learners to new words in context so that meaning and usage can be inferred (21). By analogy, LLMs acquire protein semantics and reasoning ability through context-driven exposure that grounds sequence patterns in functional and structural exemplars.

The mechanism operates in three stages. First, the adaptive context provider selects candidate contexts from the protein–natural language corpus using a dual-similarity scoring scheme, For each user query $Q$, which contains both a protein sequence and a natural-language question, we compute for every candidate protein–text pair $c_i$: (i) Amino acid sequence homology, using MMseqs2 (53), which provides a percent identity score $Sim_i^{\text{seq}} \in [0, 1]$ between the query sequence and the candidate sequence. (ii) Textual similarity, using TF–IDF representations of the candidate's descriptive text or QA pair and the query question. The final similarity score for candidate $c_i$ is a weighted

combination: $S_i = \lambda Sim_i^{\text{seq}} + (1 - \lambda)Sim_i^{\text{text}}$, with $\lambda = 0.5$ by default. In all experiments, candidates with sequence-identity scores $Sim_i^{\text{seq}} \geq 0.9999$ are explicitly masked before computing $S_i$, to avoid trivial self-matches and potential data leakage from nearly identical sequences. Second, the contextual integration module structures the top-$k$ selected examples into a coherent bilingual context. Given the selected index set $\mathcal{C}$, we represent each exemplar as a triple $(seq_i, q_i, a_i)$, ordered by decreasing $S_i$. Finally, the constructed bilingual context is combined with the query and presented to the LLM as in-context examples, enabling context-grounded interpretation and evidence integration to produce biologically meaningful responses.

## 4 EXPERIMENTS

### 4.1 SETUP

**Evaluation Datasets** We comprehensively evaluated our method using 3 text-based protein understanding datasets: ❶ ProtDescribe (75) comprises 553,052 high-quality protein–text pairs extracted from Swiss-Prot. Each instance pairs an amino-acid sequence with a single textual description obtained by concatenating four annotation fields in a fixed order: protein name, function, subcellular location, and similarity. The resulting descriptions average 40–60 tokens. ❷ Protein2Text-QA (22) comprises 209,847 open-ended question–answer pairs covering 5,574 unique proteins. Each instance consists of an amino-acid sequence, a free-form question, and a concise answer; all QAs are automatically generated from PubMed abstracts/discussion/introduction sections and presented as conversational natural-language text without fixed templates. ❸ Mol-Instructions (14) comprises 2.04 M instruction instances divided into three major sections: molecule-oriented, protein-oriented, and biomolecular-text. The protein-oriented section alone contributes 505 K instructions covering diverse tasks. Each sample is formatted as a natural-language "instruction–input–output" triplet: the input is a UniProt amino-acid sequence, and the output is a free-text answer tailored to the specific task.

**Models** All experiments are conducted under identical prompting protocols and follow the leakage-controlled setting described in Sec.3.2. We first evaluate the proposed adaptive context construction method on frozen LLMs, including Qwen2.5-3B (59), Mistral-7B-Instruct-v0.3 (8), Qwen3-14B (60), Kimi-k2 (58), and GPT-4o (42), to test few-shot and compositional reasoning capabilities, thereby mimicking the dynamics of second language acquisition. In addition, we also evaluate fine-tuned protein-oriented LLMs, including Galactica-6.7b (57), BioT5-plus-base (45), InstructProtein (64) and ProLLaMA (35), which have been explicitly trained on large-scale protein corpora. These models serve as a baseline for comparison, allowing us to examine the performance gains of our method in general-purpose frozen LLMs relative to specialized protein LLMs.

### 4.2 QUALITY OF DATASET

Figure 3 (a-f) provides a multidimensional analysis of the protein sequences included in our dataset. The collection spans a wide range of sequence lengths, from short peptides to large multi-domain proteins, and covers proteins from 4,135 species across diverse evolutionary lineages. At the family level, the dataset comprises 63,749 families and 1,115 superfamilies, ensuring representation of both well-studied proteins and rare functional groups. Additional annotations capture domain composition, catalytic activity classes, and gene ontology categories, collectively highlighting the long-tail distribution across sequence space and functional categories. This diversity ensures broad biological coverage while posing realistic challenges in inferring functions for proteins, particularly for infrequent families and underexplored functions.

Figure 3 (g,h) summarizes the distribution of tasks and token composition within the dataset. The corpus encompasses four distinct protein-QA types, with sample counts ranging from 11,693 (attribute-based QA) to 32,444 (true/false QA), thereby providing balanced coverage across multiple functional perspectives. In terms of token composition, amino-acid sequences constitute nearly 70 % of the corpus, reflecting the sequence-centric nature of protein understanding tasks and highlighting the need for models to align symbolic sequence information with natural-language context effectively.

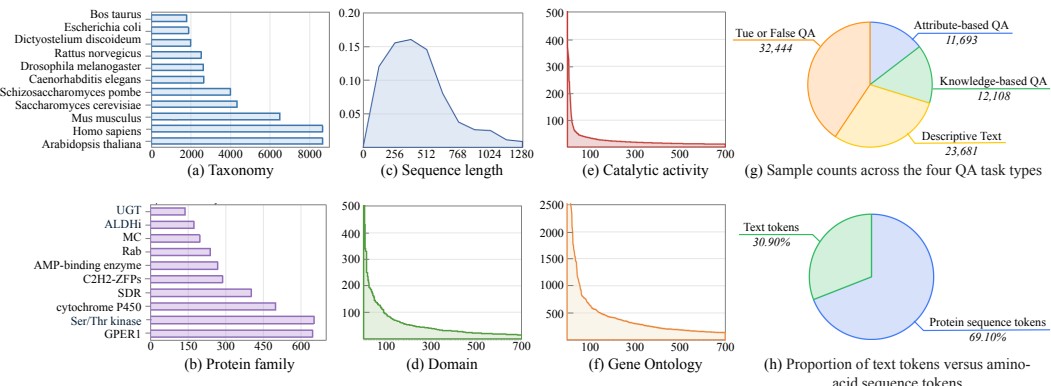

Figure 3: **Dataset statistics.** Left: Multidimensional analysis of protein amino-acid sequences, including length, domain composition, and catalytic activity. Right: Sample sizes for the four protein-QA types and the ratio of textual to amino-acid sequence tokens.

Table 1: **Comparison of different approaches on descriptive protein QA datasets** Δ *Gain* shows the percentage performance increase. ◇ indicates LLMs augmented with our adaptive context construction method. Metric: ROUGE-L (R-L), BLEU-2 (B-2), BERTScore (BS).

| Model | ProtDescribe | | | Protein2Text-QA | | | Mol-Instructions | | |
|---|---|---|---|---|---|---|---|---|---|
| | R-L | B-2 | BS | R-L | B-2 | BS | R-L | B-2 | BS |
| *Fine-tuned LLM* | | | | | | | | | |
| Galactica-6.7b (57) | 8.08 | 1.72 | 49.31 | 9.67 | 3.04 | 55.57 | 9.07 | 1.55 | 50.04 |
| BioT5+ (45) | 9.97 | 1.96 | 53.54 | 6.96 | 1.24 | 55.53 | 3.55 | 1.15 | 38.53 |
| InstructProtein (64) | 2.11 | 0.84 | 41.04 | 2.89 | 0.63 | 42.03 | 4.89 | 1.24 | 39.48 |
| ProLLaMA-7B (35) | 12.77 | 3.26 | 55.49 | 10.09 | 2.02 | 57.98 | 16.89 | 7.07 | 62.71 |
| *Frozen LLM* | | | | | | | | | |
| Qwen2.5-3B (59) | 18.45 | 7.35 | 58.05 | 23.21 | 8.64 | 68.94 | 18.54 | 6.96 | 60.91 |
| Qwen2.5-3B (59) ◇ | 26.17 | 8.02 | 61.37 | 27.19 | 12.84 | 72.11 | 22.72 | 10.65 | 64.89 |
| Δ *Gain* | **+7.72** | **+0.67** | **+3.32** | **+3.98** | **+4.20** | **+3.17** | **+4.18** | **+3.69** | **+3.98** |
| Mistral-7B-Instruct-v0.3 (8) | 14.90 | 5.70 | 58.43 | 20.97 | 9.12 | 66.01 | 17.16 | 6.33 | 59.83 |
| Mistral-7B-Instruct-v0.3 (8) ◇ | 26.35 | 10.23 | 62.66 | 22.06 | 9.88 | 69.64 | 19.40 | 7.25 | 63.60 |
| Δ *Gain* | **+11.45** | **+4.53** | **+4.23** | **+1.09** | **+0.76** | **+3.63** | **+2.24** | **+0.92** | **+3.77** |
| Qwen3-14B (60) | 23.20 | 4.47 | 60.06 | 21.02 | 8.25 | 69.44 | 14.60 | 3.68 | 60.36 |
| Qwen3-14B (60) ◇ | 32.37 | 5.68 | 63.57 | 25.49 | 12.65 | 71.53 | 20.96 | 7.53 | 65.00 |
| Δ *Gain* | **+9.17** | **+1.21** | **+3.51** | **+4.47** | **+4.40** | **+2.09** | **+6.36** | **+3.85** | **+4.64** |
| kimi-k2 (58) | 25.16 | 9.07 | 61.90 | 17.33 | 5.73 | 66.54 | 12.81 | 3.26 | 55.63 |
| kimi-k2 (58) ◇ | 32.86 | 9.12 | 64.68 | 19.10 | 6.96 | 68.16 | 18.35 | 6.04 | 64.91 |
| Δ *Gain* | **+7.70** | **+0.05** | **+2.78** | **+1.77** | **+1.23** | **+1.62** | **+5.54** | **+2.78** | **+9.28** |
| GPT-4o (42) | 18.29 | 8.07 | 60.31 | 20.84 | 8.32 | 69.52 | 17.03 | 5.62 | 61.76 |
| GPT-4o (42) ◇ | 33.29 | 12.86 | 63.91 | 26.43 | 12.86 | 72.05 | 22.90 | 8.87 | 66.31 |
| Δ *Gain* | **+15.00** | **+4.79** | **+3.60** | **+5.59** | **+4.54** | **+2.53** | **+5.87** | **+3.25** | **+4.55** |

## 4.3 MAIN RESULTS

**Accuracy gains from context-driven exposure** We evaluate our method on both descriptive QA and True/False QA protein understanding tasks. On descriptive QA datasets, our approach improves the average ROUGE-L by 6.14% across diverse open-source models and GPT-4o (42), as shown in Table 1., and human evaluation further confirms higher perceived answer quality (Figure 4). On True/False QA datasets, our method yields an additional 22.5% average accuracy improvement, as reported in Table 5. While fine-tuned protein LLMs such as InstructProtein may perform strongly on

datasets closely aligned with their training distribution, frozen general-purpose LLMs enhanced with our method remain broadly competitive across benchmarks, with performance that is comparable or superior depending on the model–task combination.

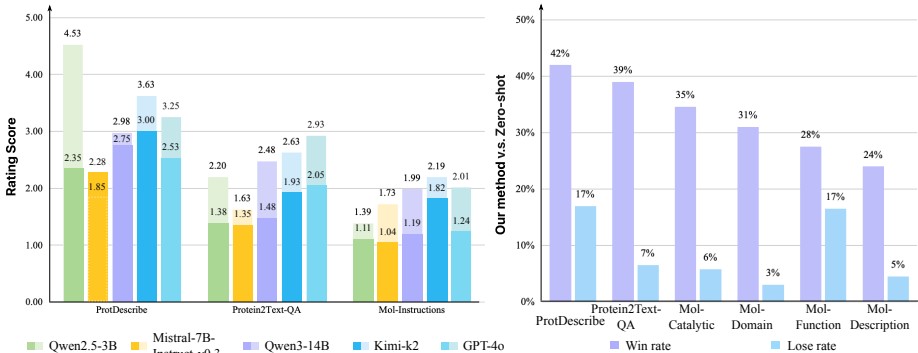

Figure 4: **Comparison of human evaluation results.** Left: Absolute human rating scores (0–5) for zero-shot model outputs (dark bars) and model outputs with adaptive context exposure (light bars) on three datasets. Right: Pairwise win/lose proportions comparing outputs with and without adaptive context exposure. Each comparison is based on 8 randomly selected cases per subset (48 cases in total across six subsets). Detailed scoring rubrics are provided in Appendix A

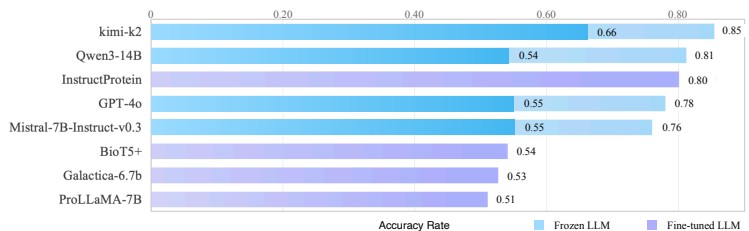

Figure 5: **Performance on True/False protein QA datasets.** Dark blue bars denote the zero-shot baseline, and light blue bars denote the corresponding models augmented with our method. The True/False QA data is directly adapted from the publicly released downstream evaluation tasks provided in InstructProtein (64).

**Performance on proteins out-of-distribution** On proteins with low similarity to the QA corpus (sequence identity $<40\%$), our method achieves a 7.12% ROUGE-L gain over the zero-shot baseline (Table 2). Zero-shot performance in this subset is on average 1.66% lower than on the full test set, likely reflecting the presence of rare or underrepresented proteins that increase task difficulty. The consistent improvement in this harder setting shows that our approach remains effective even when close homologs are absent.

**Contextual exposure vs. fine-tuned adaptation** Our contextual exposure approach surpasses fine-tuned baselines. As shown in Figure 7, it achieves an average ROUGE-L of 21.39%, outperforming the LoRA-fine-tuned (69) model (17.91%). We further compare with an analogy-based contextual-exposure method (ANALOGYKB (76)), which performs below the fine-tuned baseline in this setting. Implementation details for both methods are provided in the Appendix C. In terms of inference cost, Table 3 shows that context construction adds only 0.05 s per query, which is small relative to the 4–5 s decoding time across models.

**Varying exemplar number** ($k$) Figure 6 shows that increasing the number of exemplars $k$ improves performance up to a task-dependent optimum, after which gains diminish or reverse. The optimal $k$ differs by task. For ProtDescribe (75), which involves fixed attribute-centric questions, a larger set of bilingual exemplars from related proteins helps the model capture recurring patterns, with performance peaking at $k = 10$–$11$. In contrast, Protein2Text-QA (22) requires open-ended and

Table 2: **Performance on proteins out-of-distribution in sequence space.** Test proteins were selected by using MMseqs2 (53) to identify sequences with <40% identity to all entries in the three evaluation datasets.

| Model | ProtDescribe | | | Protein2Text-QA | | | Mol-Instructions | | |
|---|---|---|---|---|---|---|---|---|---|
| | R-L | B-2 | BS | R-L | B-2 | BS | R-L | B-2 | BS |
| Qwen2.5-3B (59) | 18.61 | 7.55 | 58.27 | 18.60 | 6.63 | 67.42 | 18.65 | 7.25 | 60.97 |
| Qwen2.5-3B (59) ◇ | 26.16 | 9.67 | 64.03 | 21.44 | 8.60 | 68.05 | 22.61 | 10.30 | 64.25 |
| Δ *Gain* | **+7.55** | **+2.12** | **+5.76** | **+2.84** | **+1.97** | **+0.63** | **+3.96** | **+3.05** | **+3.28** |
| Mistral-7B-Instruct-v0.3 (8) | 17.04 | 6.84 | 60.08 | 16.28 | 5.89 | 65.08 | 11.44 | 3.83 | 55.44 |
| Mistral-7B-Instruct-v0.3 (8) ◇ | 30.19 | 11.34 | 64.63 | 19.09 | 7.46 | 68.61 | 20.57 | 7.70 | 64.89 |
| Δ *Gain* | **+13.15** | **+4.50** | **+4.55** | **+2.81** | **+1.58** | **+3.53** | **+9.13** | **+3.87** | **+9.45** |
| Qwen3-14B (60) | 23.72 | 10.52 | 63.24 | 17.84 | 6.23 | 68.31 | 13.53 | 3.56 | 53.18 |
| Qwen3-14B (60) ◇ | 36.12 | 11.09 | 65.51 | 22.51 | 10.19 | 70.28 | 15.73 | 5.90 | 60.19 |
| Δ *Gain* | **+12.40** | **+0.57** | **+2.27** | **+4.67** | **+3.96** | **+1.97** | **+2.20** | **+2.34** | **+7.01** |
| kimi-k2 (58) | 24.41 | 9.98 | 62.79 | 13.20 | 3.22 | 64.17 | 12.74 | 3.60 | 55.05 |
| kimi-k2 (58) ◇ | 35.68 | 10.56 | 65.99 | 17.09 | 5.15 | 67.59 | 18.77 | 5.53 | 65.30 |
| Δ *Gain* | **+11.27** | **+0.58** | **+3.20** | **+3.89** | **+1.93** | **+3.42** | **+6.03** | **+1.93** | **+10.25** |
| GPT-4o (42) | 19.91 | 10.32 | 59.80 | 16.94 | 6.71 | 67.45 | 15.61 | 6.06 | 59.16 |
| GPT-4o (42) ◇ | 34.08 | 11.00 | 63.40 | 23.38 | 10.02 | 70.93 | 21.70 | 8.14 | 65.32 |
| Δ *Gain* | **+14.17** | **+0.68** | **+3.60** | **+6.44** | **+3.31** | **+3.48** | **+6.09** | **+2.08** | **+6.16** |

integrative reasoning, where fewer but relevant exemplars are beneficial; here, performance peaks earlier at $k = 3$–4. Accordingly, we adopt the task-specific optimal settings in our experiments: $k = 11$ for ProtDescribe (75), $k = 4$ for Protein2Text-QA (22), and $k = 4$ for Mol-Instructions (14).

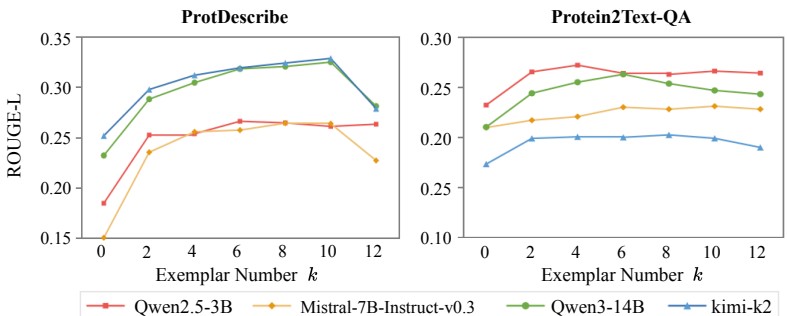

Figure 6: **Effect of varying exemplar number ($k$) on model performance.** We explored $k \in [1, 12]$ as the search space; the upper bound was set after a coarse scan up to $k = 50$ showed performance saturation around 2-12 exemplars. Metric: ROUGE-L.

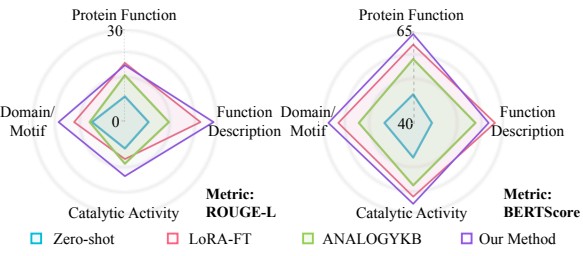

Figure 7: **Comparison of analogy-based, fine-tuned, and our methods on Mol-Instructions subtasks.** All methods are evaluated using Qwen2.5-7B (59).

Table 3: **Comparison of inference efficiency.** Times are reported in seconds (s).

| Method | Model Inference | Context Overhead |
|---|---|---|
| Zero-shot | 4.15 | – |
| LoRA-FT (69) | 2.89 | – |
| ANALOGYKB (76) | 4.94 | 0.07 |
| Our Method | 4.87 | 0.05 |

Table 4: **Ablation study of different matching strategies for exemplar selection.** Metric: BERTScore.

| Exemplar Selection | Qwen2.5-3B (59) | Mistral-7B-Instruct-v0.3 (8) | Qwen3-14B (60) | Kimi-k2 (58) | GPT-4o (42) |
|---|---|---|---|---|---|
| TF-IDF-only | 61.60 | 60.64 | 61.69 | 60.63 | 61.76 |
| MMseqs2-only | 64.45 | 62.84 | 64.88 | 64.82 | 66.12 |
| ESM2 +TF-IDF | 62.18 | 61.40 | 61.69 | 60.63 | 61.76 |
| MMseqs2 + TF-IDF | 64.89 | 63.60 | 65.00 | 64.91 | 66.31 |

**Ablation on dual-criterion context selection**    As shown in Table 4, TF-IDF–only yields the lowest scores, while MMseqs2-only performs better, indicating the usefulness of sequence-level matching. The ESM2 (30) + TF-IDF variant improves over TF-IDF but remains below MMseqs2 + TF-IDF. Overall, combining MMseqs2 with TF-IDF achieves the best performance across models.

**Case studies and qualitative evaluation**    Figure 8 illustrates that context-driven exposure produces concise, function-specific descriptions consistent with UniProt annotations. In the two examples shown, the model correctly identifies "intrinsically disordered regions", and "[4Fe-4S] RNA methyltransferase activity", whereas zero-shot outputs remain generic.

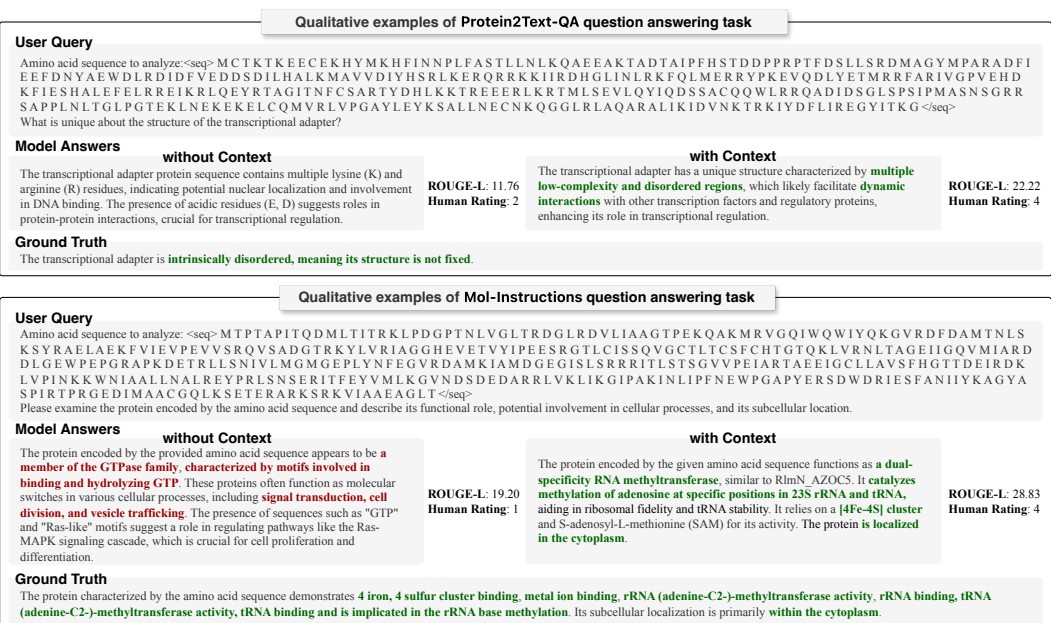

Figure 8: **Qualitative examples of protein question answering.** We present two examples with answers generated by GPT-4o (42) along with the target ground truth. The green color highlights accurate keywords, while the red color indicates prediction errors.

## 5    CONCLUSION

We have proposed the "***Protein-as-Second-Language***" framework, which leverages adaptive context construction to enhance bilingual protein understanding by dynamically integrating sequence homology and textual similarity. Supported by a dedicated ***protein–natural language bilingual dataset***, our method allows LLMs to acquire protein semantics without task-specific parameter updates. Experiments on multiple protein-language datasets demonstrate that our framework consistently outperforms zero-shot baselines, highlighting the effectiveness of context-driven learning in bridging protein sequences with functional descriptions.

## 6 ETHICS STATEMENT

This work complies with ethical standards and established research practices. All protein data were sourced from publicly available databases, with no proprietary or confidential information involved. Quality assurance and safety checks were applied to minimize harmful or inappropriate content. We acknowledge the broader risks of combining LLMs with biomolecular knowledge, including potential misuse for harmful purposes, and therefore emphasize responsible use guided by fairness, transparency, and accountability. Any harmful or unsafe applications of this dataset are strictly prohibited.

## 7 REPRODUCIBILITY STATEMENT

We provide detailed descriptions of the protein–natural language bilingual dataset (Sec. 3.1, Appendix D), the adaptive context construction mechanism (Sec. 3.2). Data processing steps and QA generation prompts for all four question types are included in Sec. 3.1 and Appendix E. Code implementing the framework and instructions for reproducing experiments on both frozen and protein-adapted LLMs will be provided as supplementary material upon acceptance.

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

## A    EVALUATION METRICS

We use the automatic metric ROUGE-L (29), BLEU-2 (43), BERTScore (78) to assess the quality of the generated text by comparing it with reference answers. For semantic similarity, we compute BERTScore (78) using SciBERT as the backbone to better capture biomedical terminology. In addition to automatic metrics, we incorporate manual checking into the evaluation pipeline and compute a human-rating score. Five evaluators with biological-research experience were asked to rate each generated answer on a 0–5 scale (the integer score corresponds to the category number minus one). All evaluators have at least two years of research experience in bioinformatics or related biological fields. The six ordinal categories they used are:

1. **Garbled** – the text is incomprehensible and lacks any readability.
2. **Inaccurate** – the text is readable but entirely incorrect and devoid of meaningful information.
3. **Partially informative** – the text offers some reference value, yet its factual correctness is poor.
4. **Moderately accurate** – roughly half of the information is correct, but several errors remain.
5. **Mostly accurate** – the content is almost entirely correct, with only minor omissions or errors.
6. **Completely correct** – the content is accurate in its entirety, without any mistakes.

## B    ADDITIONAL RESULTS

**Effect of context format**    On Protein2Text (22), we compare zero context, annotation-based context, and QA-based context. As shown in Table 5, using raw annotations reduces ROUGE-L by an average of 11.96% compared with the zero-context setting, indicating that unstructured annotations introduce noise rather than help.

**Inference efficiency and scalability**    As shown in Table 6, frozen LLMs between 3B and 14B parameters exhibit inference times of roughly 5 seconds with context, indicating limited sensitivity to model size in this range. Larger models such as Kimi-k2 (58) and GPT-4o (42) even generate slightly faster with context, likely due to decoding differences. For fine-tuned LLMs, the measured inference time appears to increase with parameter count.

Table 5: **Comparison of annotation-based and QA-based context formats.** Metric: ROUGE-L (R-L), BLEU-2 (B-2), BERTScore (BS).

| Context Format | Mistral-7B-Instruct-v0.3 (8) | | | Qwen3-14B (60) | | | Kimi-k2 (58) | | |
|---|---|---|---|---|---|---|---|---|---|
| | R-L | B-2 | BS | R-L | B-2 | BS | R-L | B-2 | BS |
| Zero-shot | 20.97 | 9.12 | 66.01 | 21.02 | 8.24 | 69.44 | 17.33 | 5.73 | 66.54 |
| Annotation-Based Context | 7.83 | 3.05 | 69.64 | 21.02 | 3.06 | 57.64 | 7.08 | 1.91 | 57.24 |
| QA-Based Context (ours) | 22.06 | 9.88 | 56.71 | 25.49 | 12.65 | 71.53 | 19.10 | 6.91 | 68.16 |

Table 6: **Comparison of inference latency across model sizes with and without context (k = 4).**

| Model | Params. | Inference without Context (s) | Inference With Context (s) |
|---|---|---|---|
| *Fine-tuned LLM* | | | |
| InstructProtein (64) | 1.3B | 1.89 | - |
| Galactica (57) | 6.7B | 2.51 | - |
| ProLLaMA (35) | 7B | 8.35 | - |
| *Frozen LLM* | | | |
| Qwen2.5-3b (59) | 3B | 1.36 | 4.92 |
| Qwen2.5-7b (59) | 3B | 4.15 | 4.94 |
| Qwen3-14b (60) | 14B | 1.96 | 5.09 |
| Kimi-k2 (58) | 1T | 8.17 | 4.65 |
| GPT-4o (42) | - | 2.34 | 1.98 |

**Structure-Level validation of LLM predictions**   To determine whether model improvements extend beyond text and reflect meaningful structural reasoning, we conducted structure validation on GPT-4o's structural descriptions for several proteins. Across the four cases shown in Figure 9, descriptions generated with contextual examples showed markedly better agreement with the structural organization predicted by AlphaFold 3 (2), including correct identification of catalytic cores, cofactor-binding regions, and multi-domain architectures. In contrast, zero-shot predictions frequently missed key structural elements, highlighting that contextual examples are essential for guiding the model toward biologically coherent structural reasoning.

**Evaluation on real-world protein scenarios**   To examine the applicability of our framework beyond benchmark datasets, we evaluated it on biologically relevant queries involving uncharacterized *Homo sapiens* proteins. For each case, a current biologically relevant question of research interest was paired with the corresponding protein amino acid sequence and input to representative LLMs guided by our framework. As illustrated in Figure 12, the models produced plausible hypotheses aligned with biological knowledge. These results demonstrate that our framework can extend to real-world scenarios, offering interpretable preliminary insights into proteins lacking experimental annotation and potentially guiding future biological investigations.

**Failure-mode analysis**   KDE comparisons of the top and bottom 25% ROUGE-L outputs show that low-performing generations are associated with lower sequence similarity $Sim_{seq}$ and slightly lower text similarity $Sim_{text}$ to their exemplars (Figure 10). The gap is most pronounced in $Sim_{seq}$, indicating that failures mainly occur when the retrieved exemplars provide insufficient semantic coverage.

## C   EXPERIMENTAL DETAILS

### C.1   FINE-TUNING SETTINGS

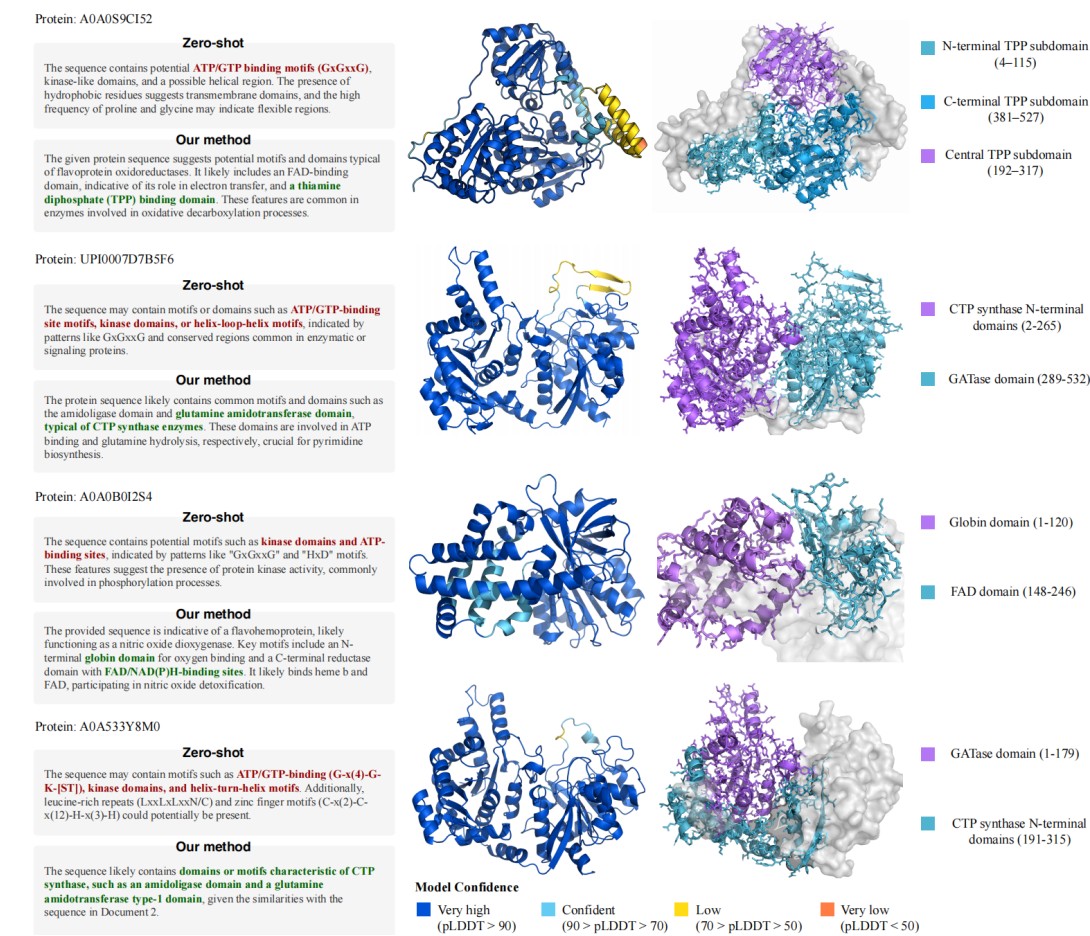

Figure 9: **Comparison of LLM structural descriptions and structure-level validation.** For each protein, the left panel shows structural descriptions generated by GPT-4o (42) with and without contextual examples. The right panel shows two AlphaFold 3 (2) visualizations of the same protein: one colored by residue-level confidence (pLDDT), and one colored by domain assignments obtained from InterProScan (47). The first two proteins (A0A0S9CI52 and UPI0007D7B5F6) have existing structural annotations in UniProt but are not included in our dataset, while the latter two (A0A0B0I2S4 and A0A533Y8M0) do not have experimentally determined or database-provided structural annotation in UniProt.

We fine-tune Qwen2.5-7B (59) on our protein–text bilingual corpus using LoRA with standard low-rank settings (rank = 8, $\alpha$ = 32, dropout = 0.05). Training is conducted on a single GPU with DeepSpeed ZeRO-2 and bf16 mixed precision. We adopt a cosine learning-rate schedule with warm-up ratio 0.01, a global learning rate of $3 \times 10^{-4}$, gradient accumulation 32, and batch size 1 per device. We fine-tune the model for 2 epochs, which provides an efficient but effective adaptation of the base model to protein-aware instruction following.

## C.2 ANALOGYKB SETUP

To mimic the relational structure used by ANALOGYKB (76), we convert each SwissProt entry into triples (sequence, relation, annotation) and group triples sharing the same annotation type into same-relation sets. Two annotation relations are considered analogous when their triples consistently express parallel biological structures across proteins—mirroring ANALOGYKB (76)'s criterion that analogous relations must support valid cross-relation analogies. This procedure yields 71 same-relation categories and 47 analogous-relation pairs, from which we sample two exemplars per query to form the analogy-driven context.

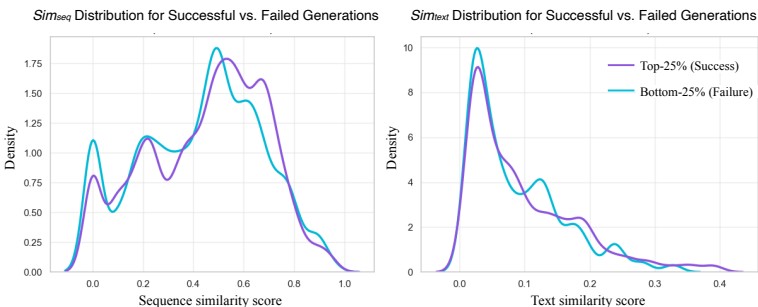

Figure 10: **KDE Distributions of Exemplar Similarity on Protein2Text.** Kernel density estimates were computed using Gaussian kernels.

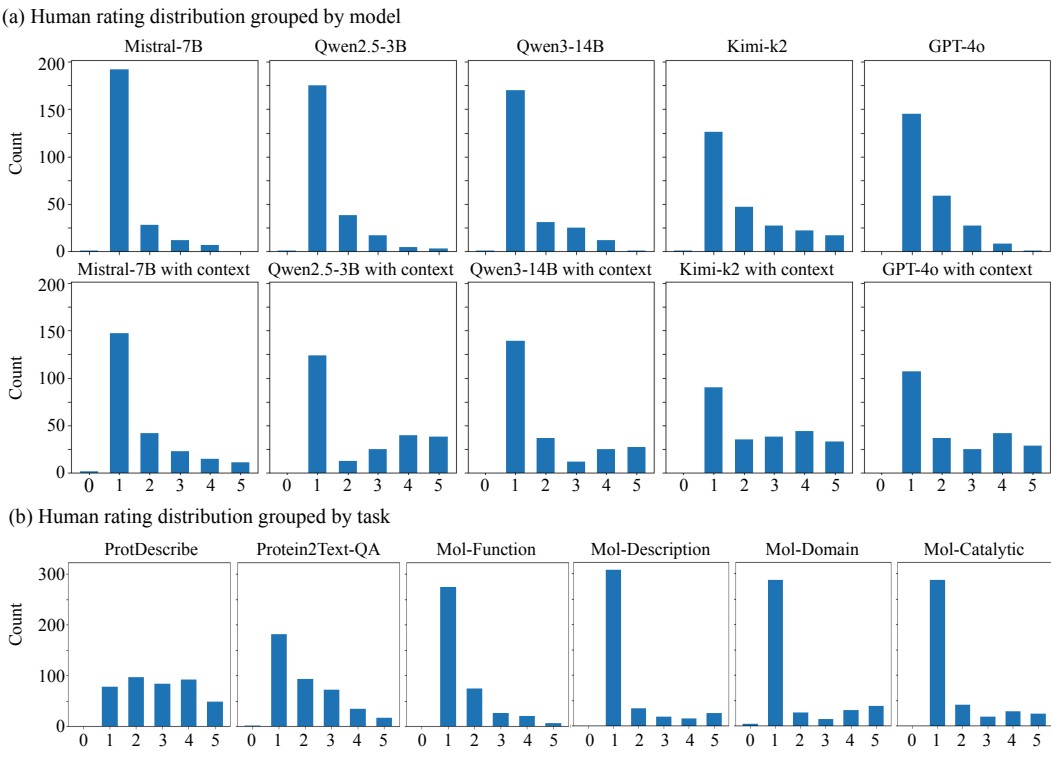

Figure 11: **Human Evaluation Score Distributions Across Models and Tasks.**

# D BILINGUAL DATASET AND QUALITY VERIFICATION

All protein entries originate from the manually curated SwissProt (4) section of UniProt, which provides high-quality annotations of protein sequences and functional descriptions. After deduplication, four types of bilingual QA pairs were generated from these entries using large language models. Examples of the four QA types are shown in Figure 14 and Figure 15. To assess the reliability of the automatically constructed corpus, we conducted a manual audit on 500 randomly sampled QA pairs from the full set of 79,926 examples. Each pair was independently reviewed by two domain experts (each with over two years of research experience in bioinformatics or related areas) along three criteria:

1. **Semantic Fidelity** – The QA must faithfully reflect the information contained in the corresponding Swiss-Prot entry without introducing unsupported claims.

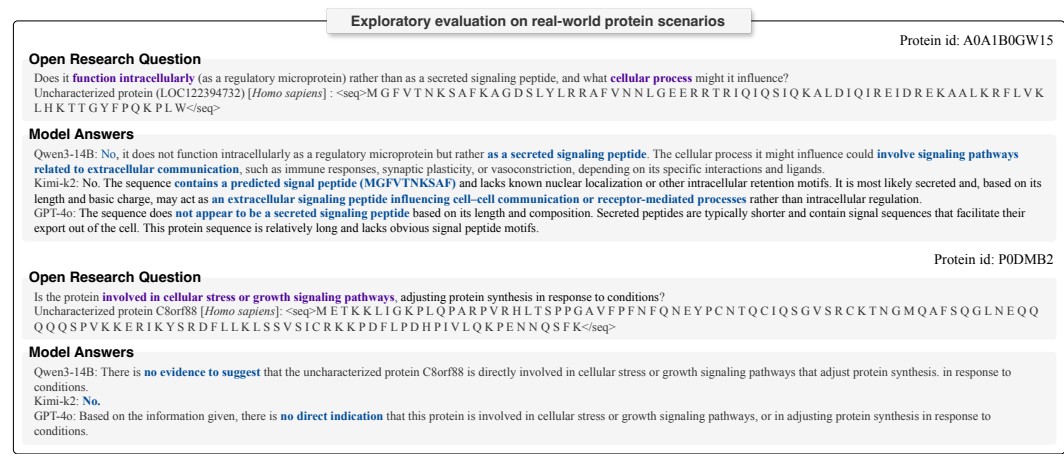

Figure 12: **Exploratory evaluation of our framework on real-world protein scenarios.**

2. **Biological Correctness** – Functional, structural, and localization statements must be biologically plausible and consistent with established knowledge.

3. **Linguistic Accuracy** – The question and answer must be grammatically correct, unambiguous, and fluent in natural language.

A QA pair was labeled pass only if it met all three criteria. Under this rubric, 95% of the sampled items passed, indicating that the bilingual dataset maintains high semantic coherence and biological reliability. Representative failure cases are shown in Figure 13.

## E    LLM STATEMENT

We acknowledge the use of LLMs in this work. Specifically, DeepSeek-R1 (17) was employed for two purposes: (i) polishing the English presentation of the manuscript, and (ii) generating bilingual dataset entries from curated protein annotations, where the prompts were carefully designed to ensure scientific accuracy and linguistic quality. Below we provide the exact prompts used for each bilingual QA type in the dataset construction process.

Prompt for `Attribute-based Answer` generation is following:

```
"Based on the provided annotations, compose a concise protein information description in the
following fixed format:
PROTEIN NAME: ...
FUNCTION: ...
SUBCELLULAR LOCATION: ...
FAMILY: ...
KEY SEQUENCE MOTIF: ... (write N/A if none).
After the fixed fields, leave one blank line and proceed to the 'Extended Information'
paragraph. In fluent, professional English, supply any additional details essential for
understanding the protein, integrating all relevant annotation content in a coherent
narrative. Maintain brevity and avoid redundancy."
```

Prompt for `True or False QA` generation is following:

```
"You are a protein science expert. Please read the UniProt entry above and design 1 True/False
question that meets all of the following rules:
(1) The stem must weave together diverse distinct knowledge dimensions from the entry (e.g.,
catalytic chemistry, structural biology, disease relevance, evolutionary conservation, PTM,
mutational effect, regulatory mechanism, substrate selectivity, experimental evidence,
GO term, PDB ID, cofactor, physiological pathway, drug-target potential).
(2) Do not include the words 'True/False' in the stem; hide the decisive technical point
within the details.
(3) Give True or False, followed by an explanation.
Use this exact output template: Stem: ...; Answer: ...; Explanation: ..."
```

Prompt for `Descriptive Text` generation is following:

```
"Based on the given annotation information of the protein, describe the given amino-acid
```

Figure 13: **Examples of Quality Issues in LLM Generated Protein QA.**

sequence in one coherent paragraph that covers:
(1) its main function and any bound cofactors/ligands,
(2) subcellular localization and the biological process involved,
(3) key domains or motifs, and
(4) the specific reaction catalyzed if it is an enzyme. The description begins with A
sentence pattern like
'A short report on the protein with the given amino acid sequence highlights:'
or 'A brief overview of the protein with the provided amino acid sequence is as follows:'
or 'A concise description of the protein with the specified amino acid sequence includes:'
or 'An outline of the key aspects of the protein with the corresponding amino acid sequence
is as follows:'
or 'A summary of the protein's main attributes with the input amino acid sequence reveals:'
(uses similar synonymous sentences to avoid uniformity)."

Prompt for `Knowledge-based QA` generation is following:

"Based on the provided annotations, generate exactly 1-9 distinct, single-sentence questions
that a researcher would naturally ask to fully interrogate this protein. Guidelines:
(1) Each question must probe a different biological dimension (expression, localization,
mechanism, regulation, phenotype, disease, evolution, interaction, structure/properties).
(2) Keep questions concise, fluent.
(3) One per line, numbering, and the corresponding answers to these questions are concise and

**Examples of attribute-based protein-QA**

Protein id: Q29DY1

**Protein**

Kinesin-like protein Klp68D: <seq>M S A K S R R P G T A S S Q T P N E C V Q V V V R C R P M S N R E R S E G S P E V V N V Y P N R G V V E L Q N V V D A N K E Q R K V F T Y D A A Y D A S A S Q T T L Y H E V V F P L V S S V L E G F N G C I F A Y G Q T G T G K T F T M E G V R G N D D L M G I I P R T F E Q I W L H I N R T E N F Q F L V D V S Y L E I Y M E E L R D L L K P N S K H L E V R E R G S G V Y V P N L H A I N C K S V D D M I R V M K V G N K N R T V G F T N M N E H S S R S H A I F M K I E M C D T E T N T I K V G K L N L I D L A G S E R Q S K T G A S A E R L K E A S K I N L A L S S L G N V I S A L A E S S P H V P Y R D S K L T R L L Q D S L G G N S K T I M I A N I G P S N Y N Y N E T L T T L R Y A S R A K S I Q N Q P I K N E D P Q D A K L K E Y Q E E I E R L K R L I A P Q Q Q Q R S E K Q G T I K K Q R V K K P K K E P I S Q E L I G S A L Q A S S A D L Q V D E D R D S D G D G A E S E S D K E N E A E V A K S N E E L E R E R V E N A K L A A K L A E L E G Q L V R G G K N L L D T Y S E R Q I E L E K K L V E I A E R K K R E I E I Q Q Q L E L Q E E T T L E I R E R N V S L E Q E V E L K K R K L S K C Y A K Y L A L Q Q E L N D C K H D H N Q D L R E L E M A Q N E L V K E L K R Q L L I I D N F V P I E V K Q R L Y T Q A K Y D E E Q E E W K F S S F P L P L P P S G G D G R Q G Y R R P V S H P Q R R R P T S E H A L Q E A K S N A P S S L R F K S E N I V S Y E L E M P C R T T Q E Y R T P K V S A S L Q A V L A Q A M Q T G G D D I D I V D S H T N S L R S R L E N I I N A N S S S N G G P G S G A G P L A A N T A G S G V G S M P N V R N I K S S R G L P S A G T A L D S N R R P P T G R I P A K K P A S A Y P K A R G L V N K</seq>

**Question**

Analyze the provided protein amino acid sequence, accurately predict and describe its function, subcellular_location, family, key_sequence_motif.\nResponse in the following exact format:\nFUNCTION: …\nSUBCELLULAR LOCATION: …\nFAMILY: …\nKEY SEQUENCE MOTIF: …\nDetailed Analysis (≤300 words): [in one brief paragraph of ≤ 300 words, describe potential catalytic mechanisms, key domains, post-translational modifications, associations with diseases, etc.]\nKeep the tone professional yet fluent; let every word earn its place.

**Answer**

FUNCTION: Plus-end directed microtubule motor involved in anterograde axonal transport, potentially moving distinct cargos in fly neurons compared to other kinesin motors.\nSUBCELLULAR LOCATION: Cytoplasm, cytoskeleton (microtubule-associated)\nFAMILY: TRAFAC class myosin-kinesin ATPase superfamily, Kinesin family, Kinesin II subfamily\nKEY SEQUENCE MOTIF: ATP-binding site (106-113)\n\n**Extended Information**\n\nKlp68D is a microtubule-associated motor protein that utilizes ATP hydrolysis to drive anterograde transport along microtubules, likely facilitating cargo movement in *Drosophila* neurons. It contains a conserved kinesin motor domain (19-344) responsible for ATP binding and microtubule interaction, along with two coiled-coil regions (350-384, 432-580) that may mediate dimerization or cargo binding. The protein exhibits multiple disordered regions and compositional biases, including basic and acidic residue clusters, which may regulate its function or interactions. Klp68D belongs to the Kinesin II subfamily, suggesting specialized roles distinct from conventional kinesin heavy chains. Its localization to the cytoskeleton underscores its involvement in microtubule-based motility processes.

Protein id: Q29J90

**Protein**

G-protein coupled receptor moody: <seq>M S D E T T G S L G D A F S P M D T P T T T I M P P P A D V D E S G F S H S L L T F A A V M T F L I M I V G I C G N L L T V V A L L K C P K V R N V A A A F I I S L C I A D L L F C A L V L P F Q G L R F V Q G T W R H G E V L C R L I P F I Q Y G N I G V K L F I A M I T I N R Y V M I T H Y S L Y N R I Y K R H W I A I M I A A C W L F S Y G M Q L P T L L G A W G R F G Y D A R L Q T C S I M S D R H G H S S K T T L F I T A F V I P C L V I I A C Y A K I F W V V H K S E Q R L K R H A T K Q N S I P N N L R P L A A A T S M P S G D G A N P S Q V P A G C R V S S D S N Y S T D V P D T T P G G A G G G A G V K Q Q P S R V A K R N E W R I T K M V L A I F L S F V I C Y L P I T I V K V A D K D V E H P S L H I F S Y I M L Y L S A C I N P I I Y V I M N K Q Y R K A Y K T V V F C Q P A A R L L M P F G K G N G A S S A A E K W K D T G L S N N H S R T I V S Q M S A G A T A T A T A T A A G T Q P Q S T S T Q G P V Q A L E L T A R V P D L I S K S S N L P L P Q P L P Q I P A G A R P S L T P P P P S V L T A T H S N G S G S Q R L P L K K N N H S Y T N S G F N S S V I S A N P S S S P S P S S S G G G I Y R P G I G S M G N G S A S I R R I T M V G D D I I L E E E E L P P T P T A S S P P Q M Q A P P P P P S S S R Q T T M N A L N T T P K T P I Y M N V D S P K R N Q S Y S E R N I P V P A R E G H D Q G V K D S Q G L P S K L M D K K K F P K D</seq>

**Question**

Analyze the provided protein amino acid sequence, accurately predict and describe its function, subcellular_location, family, key_sequence_motif.\nResponse in the following exact format:\nFUNCTION: …\nSUBCELLULAR LOCATION: …\nFAMILY: …\nKEY SEQUENCE MOTIF: …\nDetailed Analysis (≤300 words): [in one brief paragraph of ≤ 300 words, describe potential catalytic mechanisms, key domains, post-translational modifications, associations with diseases, etc.]\nKeep the tone professional yet fluent; let every word earn its place."

**Answer**

FUNCTION: Plus-end directed microtubule motor involved in anterograde axonal transport, potentially moving distinct cargos in fly neurons compared to other kinesin motors.\nSUBCELLULAR LOCATION: Cytoplasm, cytoskeleton (microtubule-associated)\nFAMILY: TRAFAC class myosin-kinesin ATPase superfamily, Kinesin family, Kinesin II subfamily\nKEY SEQUENCE MOTIF: ATP-binding site (106-113)\n\n**Extended Information**\n\nKlp68D is a microtubule-associated motor protein that utilizes ATP hydrolysis to drive anterograde transport along microtubules, likely facilitating cargo movement in *Drosophila* neurons. It contains a conserved kinesin motor domain (19-344) responsible for ATP binding and microtubule interaction, along with two coiled-coil regions (350-384, 432-580) that may mediate dimerization or cargo binding. The protein exhibits multiple disordered regions and compositional biases, including basic and acidic residue clusters, which may regulate its function or interactions. Klp68D belongs to the Kinesin II subfamily, suggesting specialized roles distinct from conventional kinesin heavy chains. Its localization to the cytoskeleton underscores its involvement in microtubule-based motility processes.

**Examples of true or false protein-QA**

Protein id: O43374

**Protein**

<seq>M A K R S S L Y I R I V E G K N L P A K D I T G S S D P Y C I V K V D N E P I I R T A T V W K T L C P F W G E E Y Q V H L P P T F H A V A F Y V M D E D A L S R D D V I G K V C L T R D T I A S H P K G F S G W A H L T E V D P D E E V Q G E I H L R L E V W P G A R A C R L R C S V L E A R D L A P K D R N G T S D P F V R V R Y K G R T R E T S I V K K S C Y P R W N E T F E F E L Q E G A M E A L C V E A W D W D L V S R N D F L F L G Q S K S R R H D E G N L G S L Q L E V R L R D E T V L P S S Y Y Q P L V H L L C H E V K L G M Q G P G Q L I P L I E E T T S T E C R Q D V A T N L L K L F L G Q G L A K D F L D L L F Q L E L S R T S E T N T L F R S N S L A S K S M E S F L K V A G M Q Y L H G V L G P I I N K V F E E K K Y V E L D P S K V E K V K D V G C S G L H R P Q T E A E V L E Q S A Q T L R A H L Q A L L S A L S R S V R A C P A V V R A T F R Q L F R R V R E R F P G A Q H E N V P F I A V T S F L C L R F F S P A I M S P K L F H L R E R H A D A R T S R T L L L L A K A V Q N V G N M D T P A S R A K E A W M E P L Q P T V R Q G V A Q L K D F I T K L V D I E E K D E L D L Q R T L S L Q A P P V K E G P L F I H R T K G K G P L M S S S F K K L Y F S L T T E A L S F A K T P S S K K S A L I K L A N I R A A E K V E E K S F G G S H V M Q V I Y T D D A G R P Q T A Y L Q C K C V N E L N Q W L S A L R K V S I N N T G L L G S Y H P G V F R G D K W S C C H Q K E K T G Q G C D K T R S R V T L Q E W N D P L D H D L E A Q L I Y R H L L G V E A M L W E R H R E L S G G A E A G T V P T S P G K V P E D S L A R L L R V L Q D L R E A H S S S P A G S P P S E P N C L L E L Q T</seq>

**Question**

Determine whether this statement about the given protein is true or false: although this calcium-binding protein translocates to the plasma membrane upon intracellular calcium elevation to inactivate Ras signaling, its pleckstrin homology domain mediates this membrane association through specific phosphoinositide interactions, which is essential for its GTPase-activating function.

**Answer**

False. The PH domain lacks phosphoinositide binding activity due to a leucine substitution at position 592, preventing it from mediating membrane association; calcium-dependent membrane binding occurs through its C2 domains instead.

Protein id: O43390

**Protein**

<seq>M A N Q V N G N A V Q L K E E E E P M D T S S V T H T E H Y K T L I E A G L P Q K V A E R L D E I F Q T G L V A Y V D L D E R A I D A L R E F N E E G A L S V L Q Q F K E S D L S H V Q N K S A F L C G V M K T Y R Q R E K Q G S K V Q E S T K G P D E A K I K A L L E R T G Y T L D V T T G Q R K Y G G P P P D S V Y S G V Q P G I G T E V F V G K I P R D L Y E D E L V P L F E K A G P I W D L R L M R G P A Y A F I T F C G K E A A Q E A V K L C D S Y E I R P G K H L Q T P S S K K P K N K T K E N I L E E F S K V T E G L V D V I L Y H Q P D D K K K N R G F C F L E Y E D H K S A A Q A R R R L M S G K V K V W G N V V T V E W A D P V E E P D P E V M A K V K V L F V R N L A T T V T E E I L E K S F S E F G K L E R V K K L K D Y A F V H F E D R G A A V K A M D E M N G K E I E G E E I E I V L A K P P D K K R K E R Q A A R Q A S R T S A Y E D Y Y Y H P P P M P P P I R G R G R G G G R G G Y G Y P P D Y Y G Y E D Y Y D D Y Y G Y D Y H D Y R G G Y E D P Y Y G Y D D G Y A V R G R G G G R G G R G A P P P R G R G A P P P R G R A G Y S Q R G A P L G P P R G S R G G R G G P A Q Q Q R G R G S R G S R G N R G G N V G G K R K A D G Y N Q P D S K R R Q T N N Q Q N W G S Q P I A Q Q P L Q Q G G D Y S G N Y G Y N N D N Q E F Y Q D T Y G Q Q W K</seq>

**Question**

Determine whether this statement about the given protein is true or false: although this protein's SUMOylation at specific lysine residues supports its role in mRNA processing complexes, and mutations causing an autosomal recessive neurodevelopmental disorder implicate its functional importance, structural studies reveal that these covalent modifications occur exclusively at lysine-359 within its third RNA recognition motif domain

**Answer**

False; Explanation: SUMOylation occurs at Lys-13, Lys-171, and Lys-359, not exclusively at Lys-359, and the associated neurodevelopmental disorder is autosomal dominant, not recessive.

Figure 14: **Illustrative examples of the four bilingual QA types (Part 1).**

---

**Examples of descriptive text protein-QA**

Protein id: Q2V4L8

**Protein**

<seq>MASPFFFVFLLSALSLENTYASPNYREALSKSLLFFQGGQRSGRLPSDQQLSWRSSSGLSDGSSAHVDLTGGYYDAGDNVKFNFPMAFTTT
MLSWSSLEYGKKMGPELQNSRVAIRWATDYLLKCARATPGKLYVGVGDPNGDHKCWERPEDMDTPRTVYSVSPSNPGSDVAAETAAALAAS
SMVFRKVDPKYSRLLLATAKKVMQFAIQYRGAYSNSLSSSVCPFYCSYSGYKDELLWGAAWLHRATNDPYYTNFIKSLGGGDQPDIFSWDN
KYAGAYVLLSRRAVLNKDNNFELYKQAAENFMCKILPNSPSSSTKYTKGGLMYKLPQSNLQYVTSITFLLTTYAKYMKSTKQTFNCGNSLIV
PNALINLSKRQVDYVLGVNPMKMSYMVGFSSNFPKRIHHRGSSLPSRAVRSNSLGCNGGFQSFRTQNPNPNILTGAIVGGPNQNDEYPDQRD
DYTRSEPATYINAAFVGPLAYFAASRSP</seq>

**Question**

In one coherent paragraph, describe the protein represented by the given amino-acid sequence, covering: (1) its main function and any bound cofactors/ligands, (2) subcellular localization and the biological process involved, (3) key domains or motifs, and (4) the specific reaction catalyzed if it is an enzyme.

**Answer**

A brief overview of the protein with the provided amino acid sequence (Q2V4L8, GUN3_ARATH) is as follows: This enzyme, an endoglucanase 3 (EC 3.2.1.4) from *Arabidopsis thaliana*, catalyzes the endohydrolysis of (1→4)-β-D-glucosidic linkages in cellulose, lichenin, and cereal β-D-glucans, facilitating polysaccharide degradation without requiring bound cofactors. It is secreted to the extracellular region, particularly localized to the plant-type cell wall and Golgi apparatus, where it participates in cell wall organization and cellulose catabolic processes. Key features include its classification within the glycosyl hydrolase 9 family, a catalytic nucleophile at position 77, additional active sites at residues 402, 453, and 462, an N-linked glycosylation site at Asn-370, and an N-terminal signal peptide (residues 1–21) for secretion. Functionally, it mediates root cap cell sloughing through cell wall breakdown, is specifically expressed in root cap cells, and is down-regulated by auxin and abscisic acid. The mature chain spans residues 22–484 and may undergo alternative splicing.

Protein id: Q39056

**Protein**

<seq>MISTLRRAVFLRRFPAVVSPIKRAFSSRIDDEFDPQIMNINELNQEMQSIFGQEPSPDGPGTMDFSELKSSKIEPLRSKNIDFRQQIEYHKS
THSSKNDSQAIEQYAKVASDMSKLTHVGIAGEAQMVDVSSKDNSKRTALACCKVILGKRVFDLVLANQMGKGDVLGVAKIAGINGAKQTSS
LIPLCHNIALTHVRVDLRLNPEDFSVDIEGEASCTGKTGVEMEAMTAVSVAGLTVYDMCKAASKDISITDVRLERKTGGKSGSWSRL</seq>

**Question**

Determine whether this statement about the given protein is true or false: although this protein's SUMOylation at specific lysine residues supports its role in mRNA processing complexes, and mutations causing an autosomal recessive neurodevelopmental disorder implicate its functional importance, structural studies reveal that these covalent modifications occur exclusively at lysine-359 within its third RNA recognition motif domain

**Answer**

A concise description of the protein with the specified amino acid sequence (Q39056, CNX3_ARATH) is as follows:\n\nThis mitochondrial protein, localized specifically within the mitochondrial matrix via an N-terminal transit peptide (residues 1-32), functions as a cyclic pyranopterin monophosphate synthase (EC 4.6.1.17). It is essential for molybdenum cofactor biosynthesis, catalyzing the specific lyase reaction that converts (8S)-3',8-cyclo-7,8-dihydroguanosine 5'-triphosphate into cyclic pyranopterin monophosphate (cPMP) and diphosphate (RHEA:49580). Key residues form an active site (including residue 240) and substrate binding sites (residues 187-189 and 225-226). Belonging to the MoaC protein family, it assembles as a homohexamer and is predominantly expressed in the roots of *Arabidopsis thaliana*. Its primary biological role is in the \"Mo-molybdopterin cofactor biosynthetic process\" (GO:0006777).

---

**Examples of knowledge-based protein-QA**

Protein id: O35854

**Protein**

<seq>MSAAILGQVWTRKLLPIPWRLCVPGRCVSSNFKAADLQVQVTREPQKKPAPSQPLLFGKTFTDHMLMVEWNSKTGWGPPRIQPFQNLTL
HPACSGLHYSLQLFEGLKAYKGRDKQVRLFRPWLNMDRMLRSARRLCLPDFDKQELLECIRQLIEVDKDWVPDGNGTSLYVRPVLIGNEPS
LGVGMVTQALLFVILCPVGSYFPGDSMTPVSLLADPSFVRAWIGGVGDCKLGGNYGPTVAVQQEAQKKGCEQVLWLYGPDHQLTEVGTMNI
FVYWTHEDGELELATPPLDGIILPGVVRQSLLDLARTWGEFRVAERKVTMKELKRALEEGRVREVFGSGTACQVCPVHQILYEGKQLHIPTM
ENGPELILRFQKELKAIQYGTSAHDWMLRV</seq>

**Question**

You are a molecular biology professor writing a protein description for undergraduates. Based on the characteristics of the given protein amino acid sequence, briefly answer the following questions: 1. What is the primary catalytic function of this protein in branched-chain amino acid metabolism?\n2. Where is this protein localized within the cell?\n3. What cofactor is essential for its enzymatic activity?\n4. How does its substrate affinity (KM) vary among the branched-chain amino acids?\n5. What is the quaternary structure of this protein?\n6. Which post-translational modifications are known to occur on this protein?\n7. What is the physiological direction of its catalytic reactions?\n8. In which tissues is this protein expressed?\n9. What family of enzymes does this protein belong to?

**Answer**

1. It catalyzes the transamination of leucine, isoleucine, and valine with 2-oxoglutarate to form their respective α-keto acids and glutamate.\n2. It is localized in the mitochondrion.\n3. Pyridoxal 5'-phosphate is the essential cofactor.\n4. The KM values are 1 mM for L-leucine and L-isoleucine, and 5 mM for L-valine.\n5. It forms a homodimer.\n6. It undergoes N6-(pyridoxal phosphate)lysine modification at position 230 and N6-acetyllysine at position 322.\n7. The reactions proceed physiologically from left to right (branched-chain amino acids to α-keto acids).\n8. It is expressed in all tissues.\n9. It belongs to the class-IV pyridoxal-phosphate-dependent aminotransferase family.

Protein id: Q9LXN8

**Protein**

<seq>MAFSMLFTGHAECVKFVKKFNLPLLVTGGGGYTKENVARCWTVETGILLDTELPNEISENDYIKYFAPDFSLKIPGGHIENLNTKSYISSI
KVQILENLRYIQHAPSVQMQEVPPDFYIPDFDEDEQNPDVRVDQRSRDKQIQRDDEYFDGDNDNDAS</seq>

**Question**

You are a molecular biology professor writing a protein description for undergraduates. Based on the characteristics of the given protein amino acid sequence, briefly answer the following questions: 1. What is the enzymatic reaction catalyzed by this protein?\n2. Where is this protein localized within the cell?\n3. What is the role of this protein in chromatin organization?\n4. Which histone proteins are substrates for its deacetylation activity?\n5. How does its activity contribute to transcriptional regulation?\n6. What are the functional implications of its disordered region (residues 120-158)?\n7. Does it form part of any multiprotein complexes?\n8. What is the significance of its basic and acidic residue compositional biases?\n9. How does it compare to other members of the histone deacetylase family?

**Answer**

1. It catalyzes the hydrolysis of N(6)-acetyl-L-lysyl-[histone] to produce L-lysyl-[histone] and acetate.\n2. It is localized in the nucleus.\n3. It is involved in chromatin organization through histone deacetylation, leading to epigenetic repression.\n4. It deacetylates lysine residues on the core histones H2A, H2B, H3, and H4.\n5. Its deacetylation activity represses transcription by modifying chromatin structure.\n6. The disordered region may contribute to protein flexibility or interaction with other molecules.\n7. Yes, it acts via the formation of large multiprotein complexes.\n8. These biases may influence protein-protein interactions or substrate binding.\n9. It belongs to the histone deacetylase family and shares similar functional domains.

Figure 15: **Illustrative examples of the four bilingual QA types (Part 2).**

```
clear.
(4) The questions can be appropriately flexible and occasionally combined with some actual
scenarios or content related to species.
The Questions and Answers are divided into two parts (wrapped with <Questions><\\Questions>
and <Answers><\\Answers> respectively). All the information in the Q&A should be based entirely
on the given annotations and should not be supplemented by yourself."
```

## F  METRICS

We evaluate semantic similarity using BERTScore (Zhang et al., 2020). To better capture biomedical terminology, we use SciBERT (allenai/scibert scivocab uncased) as the backbone model. Since BERT-based models have a maximum context window of 512 tokens, we follow common practice and truncate longer sequences to 512 subwords before computing token-level cosine similarities.

