# OpenReview forum: "Protein as a Second Language for LLMs"
_ICLR.cc/2026/Conference — Submitted to ICLR 2026_

### Official Review · Reviewer_Xf5Z · 2025-10-24

**Soundness:** 2
**Presentation:** 3
**Contribution:** 2
**Rating:** 4
**Confidence:** 4

**Summary:**

This paper introduces the "Protein-as-Second-Language" framework, which aims to enable large language models (LLMs) to interpret protein (amino acid) sequences as if they were acquiring a second symbolic language. By curating a bilingual dataset of almost 80k protein-question-answer triples and implementing an adaptive context construction mechanism, the approach supplies protein sequence–question–answer exemplars as in-context learning cues for frozen LLMs. Extensive experiments on multiple protein-text QA benchmarks demonstrate that LLMs guided by this framework substantially outperform their zero-shot counterparts, and in some cases, even domain-specialized, fine-tuned protein LLMs.

**Strengths:**

1. The paper cleverly reframes protein sequences as a "second language," allowing general-purpose LLMs to build protein-function mappings in a zero-shot regime. This paradigm bridges symbolic biological and natural languages, bypassing the need for task-specific fine-tuning or retraining and creatively leverages LLMs' in-context learning strengths.
2. A substantial and well-constructed bilingual (protein-natural language) dataset is curated, addressing redundancy at both sequence and annotation levels. Figures 1 and 3 illustrate a careful, step-wise reduction of redundancy with diverse coverage across species, protein families, superfamilies, and ontology categories.
3. The method is evaluated on multiple datasets (ProtDescribe, Protein2Text-QA, Mol-Instructions) with frozen, general-purpose LLMs (Qwen, GPT-4o, Mistral, Kimi) and compared against strong protein-LLM baselines (BioT5-plus, ProLLaMA). Main results (Table 1) and multiple figures demonstrate robust performance gains in both automatic (ROUGE-L) and human evaluations, frequently surpassing or matching fine-tuned specialized models.

**Weaknesses:**

Weaknesses
1. The methodology’s underpinnings, particularly the bilingual context construction mechanism (Section 3.2), lack precise mathematical formalization. While Equation-based thresholds (for sequence and annotation deduplication, Section 3.1.1/3.1.2) are provided, the procedure for query-to-context matching (how candidate exemplars are scored and integrated, and any ranking or aggregation formulae) is only described at a high level. For instance, does selection use hard thresholds or similarity-weighted composition? What is the precise mathematical form of the query-context similarity metric for textual and sequence components? Without a formal definition of, for example, the joint scoring function or aggregation, it is difficult for others to re-implement or to affirm the reproducibility or generalizability of the context construction process. Explicit equations or algorithms are expected for a submission at this technical level.
2. While the empirical evaluation is extensive by the standards of current protein-language model research, critical baselines are missing:
   - Direct comparison with leading analogy/reasoning-augmented LLM paradigms, such as those leveraging knowledge graphs (e.g., ANALOGYKB[1]), hierarchical retrieval (e.g., BeamAggR[2]), or reinforcement-learning-based self-correction (e.g., SeRL[3], AlphaEdit[4], Self-Correct RL[5]), is not present.
   - There is also a missed opportunity to benchmark efficiency: the computational cost of in-context "second language" understanding (which requires substantial prompt assembly with many exemplars) is never compared directly to the one-time fine-tuning cost of protein LLMs, or to parameter-efficient adaptation approaches (e.g., S-LoRA[6], MoDeGPT[7]). Without this, claims of scalability remain qualitative.
3. The central claim—LLMs can generalize protein understanding more efficiently through contextual exemplars—is only validated empirically under a restricted set of Q/A regimes. There is no attempt to analyze theoretical properties: e.g., under what assumptions does the contextual analogy paradigm guarantee generalization or compositional reasoning for proteins? What are failure modes if distributional shifts exist (out-of-distribution proteins/annotations not covered by exemplars)? The paper could benefit from at least some basic analysis or a discussion of the limitations.

references
[1] Yuan, S., Chen, J., Sun, C. (2024): "ANALOGYKB: Unlocking Analogical Reasoning of Language Models with A Million-scale Knowledge Base"
[2] Chu, Z., Chen, J., Chen, Q. (2024): "BeamAggR: Beam Aggregation Reasoning over Multi-source Knowledge for Multi-hop Question Answering"
[3] Fang, W., Liu, S., Zhou, Y. (2025): "SeRL: Self-Play Reinforcement Learning for Large Language Models with Limited Data"
[4] Fang, J., Jiang, H., Wang, K. (2025): "AlphaEdit: Null-Space Constrained Model Editing for Language Models"
[5] Kumar, A., Zhuang, V., Agarwal, R. (2025): "Training Language Models to Self-Correct via Reinforcement Learning"
[6] Wu, Y., Piao, H., Huang, L. (2025): "S-LoRA: Scalable Low-Rank Adaptation for Class Incremental Learning"
[7] Lin, C., Gao, S., Smith, J. S. (2025): "MoDeGPT: Modular Decomposition for Large Language Model Compression"

**Questions:**

1. Can the authors formalize the context construction mechanism, especially the mathematical definitions of exemplar ranking, weighting, or aggregation (e.g., is there a score function for context selection, or are choices made heuristically)? Please clarify with explicit pseudocode or equations.
2. What is the full breakdown of annotation QA pass/fail in the 5% discarded portion of the dataset? Are there any systematic biases or edge cases in the rejected data?
3. Did the authors attempt to benchmark model inference/prompt assembly time versus fine-tuned protein LLMs, or estimate resource requirements (memory/latency) for large context window assembly in practical use?
4. Could the authors compare directly with analogy-driven or multi-hop retrieval LLM frameworks (such as ANALOGYKB, BeamAggR) and efficient adaptation methods (S-LoRA, MoDeGPT)?
5. Can the authors clarify the apparent error in the human-in-the-loop Krippendorff's $\alpha$ (is 0.72% a typo?), and provide more detail on the distribution of human evaluations by task/model?

---

> ### Author Response · Authors · 2025-12-03
>
> We sincerely thank you for the thoughtful and constructive feedback, which helped us refine both the methodology and presentation.
>
> >W1 & Q1. Formalization of the context construction mechanism.
>
> We have revised Section 3.2 to explicitly define all components of the bilingual context construction pipeline.
>
> (1) Query–context scoring is now formalized.
> For each candidate $c_i$, we compute  $ Sim_i^{\text{seq}} \in [0,1] $ via MMseqs2 percent identity, and  $Sim_i^{\text{text}} = \cos(\text{TFIDF}(q_Q), \text{TFIDF}(q_i)).$ The joint similarity score is now explicitly defined as  $ S_i = \lambda\, Sim_i^{\text{seq}} + (1-\lambda)\, Sim_i^{\text{text}} $ with $\(\lambda = 0.5\)$.
>
> (2) Selection uses ranking, not hard thresholds.
> All unmasked candidates are scored, and the top-$k$ exemplars  $\mathcal{C} = \text{Top-}k(S_i) $
> are selected and ordered by decreasing $S_i$. We clarified that no thresholding is applied to the final score.
>
> (3) Integration procedure clarified.
> The revised text specifies that selected triples $(seq_i, q_i, a_i)$ are concatenated in score order to form the bilingual context before being combined with the query.
>
> >W2 & Q4. Comparison with analogy-based and fine-tuned baselines, and efficiency considerations.
>
> (1) Comparison with analogy-based and reasoning-style retrieval frameworks.
>
> We added comparisons against both analogy-driven frameworks and fine-tuned LLMs (Figure 7) and report ROUGE-L scores across four Mol-Instructions subtasks (Protein Function, Domain/Motif, Function Description, Catalytic Activity). All three methods (LoRA-FT, ANALOGYKB, and ours) were evaluated using Qwen2.5-7B [1] as the base model for consistency. The results (shown in the table below) indicate that our method outperforms both the analogy-driven baseline and the fine-tuned LoRA model, demonstrating that bilingual context construction is competitive with both types of approaches.
>
> | Method     | Function | Description | Domain | Catalytic |
> |------------|----------|-------------|--------|-----------|
> | Zero-shot  | 8.23     | 7.61        | 8.48   | 10.59     |
> | LoRA-FT    | 19.05    | 24.33       | 11.89  | 16.35     |
> | ANALOGYKB  | 15.05    | 14.17       | 13.37  | 11.33     |
> | Ours       | 18.29    | 28.55       | 17.38  | 21.35     |
>
> (2) Efficiency comparison and scalability.
> As for efficiency, context construction adds only 0.05 s per query (with $k=4$), comparable to ANALOGYKB’s 0.07 s, and small relative to the 4–5 s model inference time. LoRA-FT incurs no per-query overhead and offers faster inference, but still requires a separate fine-tuning stage, whereas our method remains fully training-free.
>
> | Method      | Model Inference (s) | Context Overhead (s) |
> |-------------|----------------------|------------------------|
> | Zero-shot   | 4.15                 | –                      |
> | LoRA-FT     | 2.89                 | –                      |
> | ANALOGYKB   | 4.94                 | 0.07                   |
> | Our Method  | 4.87                 | 0.05                   |
>
> These comparisons have been added to Figure 7 and Table 3 in the revised manuscript.
>
> >W3. On theoretical assumptions and failure modes.
>
> We appreciate the reviewer’s request for a deeper examination of theoretical properties and potential failure modes. While a full formal theory of contextual analogy for protein understanding is beyond the scope of this work, we have added an empirical failure-mode analysis in Appendix B (Figure 10). By comparing the KDE distributions of exemplar similarities for the top and bottom 25% ROUGE-L outputs, we find that low-performing generations consistently receive exemplars with lower sequence similarity and slightly lower text similarity. This suggests that failures under distributional shift often coincide with limited exemplar coverage, and we now explicitly discuss this limitation in the revised manuscript.
>
> [1] Yang, A., Yu, B., Li, C., Liu, D., Huang, F., Huang, H., Jiang, J., Tu, J., Zhang, J., Zhou, J. and Lin, J., 2025. Qwen2. 5-1m technical report. arxiv preprint arxiv:2501.15383.

---

> ### Author Response · Authors · 2025-12-03
>
> >Q2. Breakdown of the 5% rejected QA pairs and potential biases.
>
> Among the 5% discarded QA pairs, 56% of issues involve semantic fidelity, 48% involve biological inaccuracies, and 20% reflect linguistic problems, with 16% containing multiple error types. we observe the error distribution summarized in the table below. Error patterns vary across QA formats: Knowledge-based QA contributes the largest share of errors across all three dimensions, followed by Descriptive Text QA. Attribute-based QA exhibits only occasional minor issues, while True/False QA shows relatively few problems overall. We did not identify any noticeable or systematic bias toward particular protein families or annotation categories.
>
> | QA Type             | Semantic Fidelity | Biological Correctness | Linguistic Accuracy |
> |---------------------|-------------------|-------------------------|----------------------|
> | Attribute-based QA  | 0.14              | 0.08                    | 0                 |
> | Knowledge-based QA  | 0.36              | 0.42                    | 0.20                 |
> | Descriptive Text QA | 0.29              | 0.33                    | 0.60                 |
> | True or False QA    | 0.21              | 0.17                    | 0.20                 |
>
>
> >Q3. Inference/prompt-assembly cost and practical latency.
>
> The context-assembly overhead is only 0.05 s per query (with $k=4$). As shown in the table below, frozen LLMs equipped with our contextual exemplars maintain inference times around 5 seconds, while fine-tuned protein LLMs generally exhibit increasing latency as parameter size grows.
>
>
> | Model            | Params. | Inference without Context (s) | Inference With Context (s) |
> |------------------|---------|-------------------------------|-----------------------------|
> | **Fine-tuned LLM** |         |                               |                             |
> | InstructProtein  | 1.3B    | 1.89                          | -                           |
> | Galactica        | 6.7B    | 2.51                          | -                           |
> | ProLLaMA         | 7B      | 8.35                          | -                           |
> | **Frozen LLM**     |         |                               |                             |
> | Qwen2.5-3B       | 3B      | 1.36                          | 4.92                        |
> | Qwen2.5-7B       | 3B      | 4.15                          | 4.94                        |
> | Qwen3-14B        | 14B     | 1.96                          | 5.09                        |
> | Kimi-k2          | 1T      | 8.17                          | 4.65                        |
> | GPT-4o           | –       | 2.34                          | 1.98                        |
>
> >Q5. On Krippendorff’s α and human-evaluation details.
>
> The reported Krippendorff’s α was indeed 0.72 (not 0.72%), and we have corrected the notation in the revised manuscript. We have also added the requested details on score distributions across tasks and models in Figure 11 of the Appendix.

---

### Official Review · Reviewer_1bpb · 2025-10-26

**Soundness:** 3
**Presentation:** 3
**Contribution:** 3
**Rating:** 4
**Confidence:** 5

**Summary:**

This paper proposes Protein-as-Second-Language (PSL), a training-free framework that enables large language models to interpret protein sequences as a “second language.” Instead of fine-tuning, PSL performs retrieval-based in-context learning by constructing bilingual contexts that pair amino-acid sequences with natural-language descriptions. The authors build a 79K protein–QA corpus via Gene Ontology–based functional grouping, MMseqs2 clustering with semantic deduplication, and automatic QA generation using DeepSeek-R1 across four question types. During inference, PSL selects relevant examples based on sequence homology and semantic similarity, forming adaptive prompts for frozen LLMs (GPT-4o, Qwen, Mistral). Across three benchmarks (ProtDescribe, Protein2Text-QA, Mol-Instructions), PSL achieves up to 17.2% ROUGE-L improvement, outperforming domain-specific models like ProLLaMA-7B and BioT5+, and reframes protein understanding as retrieval-driven bilingual reasoning rather than supervised fine-tuning.

**Strengths:**

This paper introduces a conceptually novel and computationally efficient framework that enables large language models to understand protein sequences through bilingual contextual reasoning without any fine-tuning. In addition to the framework, the authors construct a large-scale bilingual protein–text corpus containing 79,926 sequence–question–answer pairs, which serves as the foundation for retrieval-based in-context learning and systematic evaluation.

**Weaknesses:**

1. The bilingual corpus is constructed using Swiss-Prot as the primary data source, while the evaluation datasets are also derived from or highly overlap with Swiss-Prot. The paper does not provide sufficient details on how potential data leakage or overlap was prevented, which raises concerns about the fairness of evaluation.

2. Each inference involves a retrieval step to construct query-specific contexts, but the computational overhead and latency introduced by this process are not analyzed. The practical efficiency of the framework therefore remains unclear.

3. The method assumes that proteins with high MMseqs2 similarity share similar functional or semantic contexts. However, this assumption may not always hold, especially for multi-domain proteins. A more critical discussion or ablation on this assumption would strengthen the justification.

4. The experimental comparison includes only two domain-specific baselines, ProLLaMA-7B and BioT5+, which may not be sufficient to establish broad effectiveness. Including more diverse or fine-tuned protein LLMs could improve the reliability of the conclusion.

5. The framework appears to treat protein sequences and text jointly as input without a dedicated modality projector or alignment module. While this simplifies the design, it may not fully exploit cross-modal complementarities, and more structured feature integration could further enhance performance.

**Questions:**

1.Could the authors clarify whether the constructed bilingual corpus overlaps with the evaluation datasets? Since both the corpus and the benchmarks seem to originate from Swiss-Prot or related sources, it would be important to specify how potential data leakage was prevented to ensure fair evaluation.

2.The paper transforms Swiss-Prot annotations into multiple QA formats rather than using the full annotations directly. What is the motivation for this choice, and would incorporating broader and more complete biological knowledge lead to more stable contextual enhancement?

3.The proposed framework involves a retrieval step for each query to build adaptive bilingual contexts. Could the authors discuss the computational overhead introduced by this process and its impact on inference time and scalability compared with fine-tuned models?

4.The exemplar selection process is described as combining both sequence homology (via MMseqs2) and semantic similarity between QA pairs. Could the authors elaborate on how these two signals are integrated into the final retrieval score or ranking criterion?

5.The method assumes that proteins with high MMseqs2 similarity share similar contexts. Have the authors considered using alternative similarity measures, such as embedding-based similarity from ESM or structure-based similarity from ProTrek, and could they provide related ablation results?

6.In the comparison with ProLLaMA and BioT5+, were these models fine-tuned on any part of the proposed corpus (like RAFT), or were their publicly released parameters used directly for inference? Please clarify whether additional training or adaptation was performed to ensure a fair comparison.

---

> ### Author Response · Authors · 2025-12-03
>
> We sincerely thank the reviewer for the thoughtful and constructive feedback. Below we address each concern in detail.
>
> >W1&Q1. Potential data leakage between the bilingual corpus and evaluation datasets.
>
> We appreciate the reviewer’s concern regarding possible overlap between the bilingual corpus and the evaluation benchmarks, given their shared origin from Swiss-Prot. To address this, we reran all experiments under a strict leakage-prevention setting. During context construction, we exclude any protein whose MMseqs2 sequence identity to the query is ≥ 0.9999, ensuring that no near-duplicate or trivially overlapping Swiss-Prot entries can enter the exemplar set. This procedure has been added to the Section 3.2.
>
> The table below reports only the ROUGE-L metric for readability. Here, the ◇ symbol denotes LLMs augmented with our adaptive bilingual context construction method. All full experimental results (including BLEU-2 and BERTScore) have been updated in the revised manuscript. Under the strict leakage-controlled setting, the average performance gain is 6.14%, slightly lower than the previously reported ~7%, but the overall conclusions remain unchanged.
>
> | Model                      | ProtDescribe | Protein2Text | Mol-Instructions |
> |---------------------------|------------------|------------------|--------------|
> | Qwen2.5-3B                | 18.45            | 23.21            | 18.54        |
> | Qwen2.5-3B ◇              | 26.17            | 27.19            | 22.72        |
> | △ Gain                   | +7.72            | +3.98            | +4.18        |
> | Mistral-7B-Instruct-v0.3  | 14.90            | 20.97            | 17.16        |
> | Mistral-7B-Instruct-v0.3 ◇| 26.35            | 22.06            | 19.40        |
> | △ Gain                   | +11.45           | +1.09            | +2.24        |
> | Qwen3-14B                 | 23.20            | 21.02            | 14.60        |
> | Qwen3-14B ◇              | 32.37            | 25.49            | 20.96        |
> | △ Gain                   | +9.17            | +4.47            | +6.36        |
> | kimi-k2                   | 25.16            | 17.33            | 12.81        |
> | kimi-k2 ◇                | 32.86            | 19.10            | 18.35        |
> | △ Gain                   | +7.70            | +1.77            | +5.54        |
> | GPT-4o                    | 18.29            | 20.84            | 17.03        |
> | GPT-4o ◇                 | 33.29            | 26.43            | 22.90        |
> | △ Gain                   | +15.00           | +5.59            | +5.87        |
>
>
> >W2. Computational overhead.
>
> To clarify the practical efficiency of our framework, we report the context construction overhead and inference time measured on the Qwen2.5-7B model. As shown in the table below, constructing the adaptive bilingual context adds only 0.05 s per query (with $k = 4$).
>
> | Method     | Model Inference (s) | Context Overhead (s) |
> |------------|----------------------|------------------------|
> | Zero-shot  | 4.15                 | –                      |
> | Our Method | 4.87                 | 0.05                   |
>
> >W3. Assumption on MMseqs2 similarity and multi-domain proteins.
>
> We agree that sequence similarity alone does not always capture functional context, particularly for multi-domain proteins. Our framework therefore does not rely solely on MMseqs2. As shown in the ablation results in the table below, a text-only strategy underperforms a sequence-only strategy, but combining both signals consistently yields the best results across models. This indicates that semantic similarity compensates for sequence-level limitations, and that the joint MMseqs2 + TF-IDF approach provides the most reliable exemplar selection.
>
> | Exemplar Selection   | Qwen2.5-3B | Mistral-7B-Instruct-v0.3 | Qwen3-14B | Kimi-k2 | GPT-4o |
> |----------------------|------------|----------------------------|-----------|---------|--------|
> | TF-IDF-only          | 61.60      | 60.64                      | 61.69     | 60.63   | 61.76  |
> | MMseqs2-only         | 64.45      | 62.84                      | 64.88     | 64.82   | 66.12  |
> | MMseqs2 + TF-IDF     | 64.89      | 63.60                      | 65.00     | 64.91   | 66.31  |

---

> ### Author Response · Authors · 2025-12-03
>
> >W4. Baseline diversity.
>
> We have expanded the experimental comparison by adding two additional biology-capable models, Galactica-6.7B [1] and InstructProtein [2], to Table 1 and Figure 5 of the revised manuscript. These expanded baselines provide broader coverage, and the overall conclusions remain unchanged.
>
> | Model                     | ProtDescribe R-L | Protein2Text R-L | Mol-Instructions R-L |
> |---------------------------|------------------|-------------------|------------------------|
> | **Fine-tuned LLM**        |                  |                   |                        |
> | Galactica-6.7B            | 8.08             | 9.67              | 9.07                   |
> | BioT5+                    | 9.97             | 6.96              | 3.55                   |
> | InstructProtein           | 2.11             | 2.89              | 4.89                   |
> | ProLLaMA-7B               | 12.77            | 10.09             | 16.89                  |
> | **Frozen LLM**            |                  |                   |                        |
> | Qwen2.5-3B                | 18.45            | 23.21             | 18.54                  |
> | Qwen2.5-3B ◇              | 26.17            | 27.19             | 22.72                  |
> | △ Gain                   | +7.72            | +3.98             | +4.18                  |
> | Mistral-7B-Instruct-v0.3  | 14.90            | 20.97             | 17.16                  |
> | Mistral-7B-Instruct-v0.3 ◇| 26.35            | 22.06             | 19.40                  |
> | △ Gain                   | +11.45           | +1.09             | +2.24                  |
> | Qwen3-14B                 | 23.20            | 21.02             | 14.60                  |
> | Qwen3-14B ◇              | 32.37            | 25.49             | 20.96                  |
> | △ Gain                   | +9.17            | +4.47             | +6.36                  |
> | kimi-k2                   | 25.16            | 17.33             | 12.81                  |
> | kimi-k2 ◇                | 32.86            | 19.10             | 18.35                  |
> | △ Gain                   | +7.70            | +1.77             | +5.54                  |
> | GPT-4o                    | 18.29            | 20.84             | 17.03                  |
> | GPT-4o ◇                 | 33.29            | 26.43             | 22.90                  |
> | △ Gain                   | +15.00           | +5.59             | +5.87                  |
>
> >W5. Lack of a dedicated modality alignment module.
>
> Our framework is intentionally training-free, so we avoid introducing additional modality projectors or alignment modules that would require parameter updates. Instead, we rely on contextual exposure to integrate sequence and textual information at inference time. This design choice is discussed in the Section 1 (fourth paragraph), where we explain that enabling protein understanding without any model training is a core objective of our approach.
>
> >Q2. Motivation for using QA-formatted context instead of raw annotations.
>
> Our motivation is to provide models with task-aligned, question-relevant context rather than long, heterogeneous Swiss-Prot annotations. To validate this choice, we conducted an empirical comparison in which matched proteins were provided either with their raw annotations or with our QA-formatted context. As shown in the table below, using raw annotations reduces ROUGE-L by an average of 11.96% compared with the zero-context setting, indicating that unstructured annotations introduce noise rather than help.
>
> | Context Format            | Mistral-7B-Instruct-v0.3  ROUGE-L | Mistral-7B-Instruct-v0.3 BLEU-2 | Mistral-7B-Instruct-v0.3 BERTScore | Qwen3-14B ROUGE-L | Qwen3-14B BLEU-2 | Qwen3-14B BERTScore | Kimi-k2 ROUGE-L | Kimi-k2 BLEU-2 | Kimi-k2 BERTScore |
> |---------------------------|------------------------------|-------------------------------|------------------------------|---------------|----------------|--------------|--------------|--------------|-------------|
> | Zero-shot                 | 20.97                        | 9.12                          | 66.01                        | 21.02         | 8.24           | 69.44        | 17.33        | 5.73         | 66.54       |
> | Annotation-Based Context  | 7.83                         | 3.05                          | 69.64                        | 21.02         | 3.06           | 57.64        | 7.08         | 1.91         | 57.24       |
> | QA-Based Context (ours)   | 22.06                        | 9.88                          | 56.71                        | 25.49         | 12.65          | 71.53        | 19.10        | 6.91         | 68.16       |
>
>
> [1] Taylor, R., et al, 2022. Galactica: A large language model for science. arxiv preprint arxiv:2211.09085.
>
> [2] Wang, Z., et al, 2024, August. Instructprotein: Aligning human and protein language via knowledge instruction. In Proceedings of the 62nd Annual Meeting of the Association for Computational Linguistics.

---

> > ### Author Response · Authors · 2025-12-03
> >
> > >Q3. Computational overhead and scalability.
> >
> > We quantified the cost of building adaptive contexts. With k=4, the context-construction step adds only 0.05 s per query, which is negligible relative to the model’s inference time. As shown in the table below, frozen LLMs from 3B to 14B parameters maintain inference times of roughly 5 seconds with context, indicating that runtime remains stable across model sizes. In contrast, the measured latency of fine-tuned LLMs tends to increase with parameter count.
> >
> > | Model               | Params. | Inference without Context (s) | Inference With Context (s) |
> > |---------------------|---------|-------------------------------|-----------------------------|
> > | **Fine-tuned LLM**  |         |                               |                             |
> > | InstructProtein     | 1.3B    | 1.89                          | -                           |
> > | Galactica           | 6.7B    | 2.51                          | -                           |
> > | ProLLaMA            | 7B      | 8.35                          | -                           |
> > | **Frozen LLM**      |         |                               |                             |
> > | Qwen2.5-3B          | 3B      | 1.36                          | 4.92                        |
> > | Qwen2.5-7B          | 3B      | 4.15                          | 4.94                        |
> > | Qwen3-14B           | 14B     | 1.96                          | 5.09                        |
> > | Kimi-k2             | 1T      | 8.17                          | 4.65                        |
> > | GPT-4o              | –       | 2.34                          | 1.98                        |
> >
> > >Q4. Integration of sequence homology and semantic similarity.
> >
> > As detailed in the revised Section 3.2, each candidate exemplar $c_i$ receives two similarity scores: an MMseqs2 sequence-identity score $Sim_i^{\text{seq}}$ and a TF–IDF cosine similarity score $Sim_i^{\text{text}}$. These are integrated through a weighted combination: $S_i = \lambda\, Sim_i^{\text{seq}} + (1 - \lambda)\, Sim_i^{\text{text}},$ with $\lambda = 0.5$ unless otherwise noted. All unmasked candidates are ranked by $S_i$, and the top-$k$ exemplars constitute the final context.
> >
> > >Q5. Alternative similarity measures.
> >
> > We compared using ESM2-based sequence similarity versus MMseqs2-based similarity, each combined with text similarity. As shown in the table below (Table 4 in the revised manuscript), the MMseqs2 variant achieves a 3.27% higher average BERTScore, indicating that it provides more effective matching in our setting.
> >
> > | Exemplar Selection   | Qwen2.5-3B BERTScore | Mistral-7B-Instruct-v0.3  BERTScore| Qwen3-14B BERTScore | Kimi-k2 BERTScore | GPT-4o BERTScore |
> > |----------------------|------------|----------------------------|-----------|---------|--------|
> > | ESM2 + TF-IDF        | 62.18      | 61.40                      | 61.69     | 60.63   | 61.76  |
> > | MMseqs2 + TF-IDF     | 64.89      | 63.60                      | 65.00     | 64.91   | 66.31  |
> >
> > >Q6. Use of fine-tuned baselines.
> >
> > We confirm that neither ProLLaMA nor BioT5+ was fine-tuned on any part of our proposed corpus. All comparisons use their publicly released checkpoints directly for inference. Both models were originally trained on tasks closely aligned with our evaluation settings, so no additional adaptation was required to ensure fairness in comparison.

---

### Official Review · Reviewer_z9Dx · 2025-10-27

**Soundness:** 2
**Presentation:** 3
**Contribution:** 2
**Rating:** 2
**Confidence:** 4

**Summary:**

This work introduces a novel question-answering (QA) dataset focused on protein expression, localization, mechanism, and interaction. The authors also propose a retrieval-based framework that enables pretrained, generic large language models (LLMs) to analyze unknown protein sequences using an in-context learning approach. By including similar proteins and their corresponding descriptions in the prompt, the paper reports an average 7% improvement in the ROUGE-L score on QA tasks for the target unknown protein.

**Strengths:**

* This work proposes a framework that eliminates the need to train or fine-tune a task-specific LLM for protein sequence analysis.

* The paper employs dual criteria to retrieve similar proteins for augmenting the prompt, considering both sequence and text similarity.

**Weaknesses:**

* The overall novelty of the paper appears limited.  Prior works, particularly ProtEx from Google DeepMind, already explored the possibility of in-context learning for biological entity analysis. This submission however, neither mentions nor compares with them.

* The similarity-based retrieval likely limits the generalization ability of the model on unseen protein sequences that are very different from existing ones in the database. A 70% threshold is applied on sequence similarity when constructing the dataset and retrieving for the prompt, but novel or orphan proteins often share <40% identity to the existing database.

**Questions:**

Please address my two major concerns in the “Weaknesses” section first. I will reassess after the rebuttal. Other miscellaneous questions are as follows:

* Were the quality and correctness of the augmented QA from DeepSeek-R1 verified somehow? Teacher LLMs are known to hallucinate, especially on complex scientific topics like biology.

* Are GO annotations an effective criterion for grouping and redundancy reduction? As far as I understand, a group of proteins might be very different from each other even though they share similar GO annotations. This is especially the case when some proteins are not very well annotated and have very few GO annotations.

* The automatic evaluation solely relying on the ROUGE-L score is most likely not sufficient. Other metrics like accuracy (particularly for True or False QA), BLEU score, and BERTScore might provide a better understanding of the improvement in performance.

* Besides, a human evaluation is conducted in the paper, but the domain knowledge or expertise of these evaluators is not mentioned (maybe I missed it), making the reliability of this evaluation questionable.

* The term “zero-shot” used multiple times in the paper is a bit misleading, because retrieved similar proteins in the prompt may provide contextual information. The framework described in the paper is more likely a “few-shot” one.

* Numbers in Figure 4 (left) are not consistent with the bars. I would recommend a thorough double-check of all the numerical results presented in the paper.

---

> ### Author Response · Authors · 2025-12-03
>
> We sincerely thank the reviewer for the thoughtful and constructive assessment of our work, as well as for highlighting the key issues that require clarification.
>
> >W1. Limited novelty; missing discussion of ProtEx and related in-context work.
>
> We have added an “In-Context Protein Learning” subsection to Related Work (Section 2.3), where ProtEx and similar approaches are discussed together with their differences from ours. Our method introduces a bilingual sequence–text context construction framework for open-ended protein QA, allowing models to integrate natural-language evidence when forming answers. ProtEx, in contrast, focuses on label-conditioned function prediction (EC/GO/Pfam) using sets of positive and negative exemplar sequences and produces classification outputs. Because the tasks, and output formats differ fundamentally, a direct comparison to ProtEx is not applicable.
>
> >W2. Generalization on low-similarity or unseen proteins.
>
> We evaluate our method on proteins with <40% sequence identity to the QA corpus. As shown in the table below (the symbol ◇ denotes models augmented with our method), our method still yields a 7.12% ROUGE-L improvement, indicating that it remains effective even when no close homologs exist.
>
> | Model                     | ProtDescribe ROUGE-L | ProtDescribe BLEU-2 | ProtDescribe BERTScore | Protein2Text ROUGE-L | Protein2Text BLEU-2 | Protein2Text BERTScore | Mol-Instructions ROUGE-L | Mol-Instructions BLEU-2 | Mol-Instructions BERTScore |
> |---------------------------|------------------|-------------------|------------------|-------------------|-------------------|------------------|---------------|---------------|--------------|
> | Qwen2.5-3B                | 18.61            | 7.55              | 58.27            | 18.60             | 6.63              | 67.42            | 18.65         | 7.25          | 60.97        |
> | Qwen2.5-3B ◇              | 26.16            | 9.67              | 64.03            | 21.44             | 8.60              | 68.05            | 22.61         | 10.30         | 64.25        |
> | △ Gain                   | +7.55            | +2.12             | +5.76             | +2.84             | +1.97             | +0.63             | +3.96         | +3.05         | +3.28        |
> | Mistral-7B-Instruct-v0.3  | 17.04            | 6.84              | 60.08            | 16.28             | 5.89              | 65.08            | 11.44         | 3.83          | 55.44        |
> | Mistral-7B-Instruct-v0.3 ◇| 30.19            | 11.34             | 64.63            | 19.09             | 7.46              | 68.61            | 20.57         | 7.70          | 64.89        |
> | △ Gain                   | +13.15           | +4.50             | +4.55             | +2.81             | +1.58             | +3.53             | +9.13         | +3.87         | +9.45        |
> | Qwen3-14B                 | 23.72            | 10.52             | 63.24            | 17.84             | 6.23              | 68.31            | 13.53         | 3.56          | 53.18        |
> | Qwen3-14B ◇              | 36.12            | 11.09             | 65.51            | 22.51             | 10.19             | 70.28            | 15.73         | 5.90          | 60.19        |
> | △ Gain                   | +12.40           | +0.57             | +2.27             | +4.67             | +3.96             | +1.97             | +2.20         | +2.34         | +7.01        |
> | kimi-k2                   | 24.41            | 9.98              | 62.79            | 13.20             | 3.22              | 64.17            | 12.74         | 3.60          | 55.05        |
> | kimi-k2 ◇                | 35.68            | 10.56             | 65.99            | 17.09             | 5.15              | 67.59            | 18.77         | 5.53          | 65.30        |
> | △ Gain                   | +11.27           | +0.58             | +3.20             | +3.89             | +1.93             | +3.42             | +6.03         | +1.93         | +10.25       |
> | GPT-4o                    | 19.91            | 10.32             | 59.80            | 16.94             | 6.71              | 67.45            | 15.61         | 6.06          | 59.16        |
> | GPT-4o ◇                 | 34.08            | 11.00             | 63.40            | 23.38             | 10.02             | 70.93            | 21.70         | 8.14          | 65.32        |
> | △ Gain                   | +14.17           | +0.68             | +3.60             | +6.44             | +3.31             | +3.48             | +6.09         | +2.08         | +6.16        |

---

> ### Author Response · Authors · 2025-12-03
>
> >Q1. Quality of augmented QA from DeepSeek-R1.
>
> We agree that teacher LLMs may introduce hallucinations. Although a small amount of noise is unavoidable, our experiments indicate that it does not have a noticeable impact on the overall performance of the framework. We verify data quality through a manual audit of 500 randomly sampled QA pairs, where two domain experts independently evaluate each item using three criteria: semantic fidelity, biological correctness, and linguistic accuracy. An item is marked as passing only if it satisfies all three criteria (details are provided in Appendix D). Figure 13 presents representative examples of QA pairs that do not pass this quality check. Under this evaluation protocol, 95% of sampled items pass, demonstrating that the curated corpus maintains high reliability.
>
> >Q2. Use of GO annotations for grouping and redundancy reduction.
>
> (1)GO as a functional grouping signal.
>
> We acknowledge that proteins sharing GO terms may still differ in fine-grained function. Our method does not assume functional equivalence based on GO alone; instead, GO serves as a coarse, interpretable anchor that captures major semantic aspects of a protein’s description (e.g., catalytic type, molecular role, localization). Although not perfect, GO provides a widely used, structured vocabulary that is sufficient for coarse-level organization.
>
> (2) GO as an effective cue for redundancy control.
>
> Our goal is not to enforce functional identity, but to prevent duplicated QA content. Proteins with substantial GO overlap typically exhibit highly similar natural-language descriptions, which makes GO an effective indicator of potential semantic redundancy even when functional details diverge. For proteins with sparse annotations, fewer QA pairs can be generated, and thus they do not materially contribute to redundancy. In practice, the resulting dataset remains diverse while avoiding over-representation of near-identical descriptions.
>
> >Q3. Evaluation beyond ROUGE-L.
>
> In the revised version, we report additional metrics including True/False QA accuracy (Figure 4), BLEU-2, and BERTScore (Table 1, Table 2). Across all evaluated LLMs, our method improves consistently on all metrics, providing a more complete assessment than ROUGE-L alone.
>
> The table below reports the True/False QA accuracy results. The True/False QA data are directly adapted from the publicly released downstream evaluation tasks provided in InstructProtein [1]. The symbol ◇ denotes models augmented with our method. On average, our approach yields a 22.5% improvement across all evaluated LLMs.
>
> | Model                         | Accuracy |
> |------------------------------|----------|
> | ProLLaMA-7B                  | 51.05%   |
> | Galactica-6.7B               | 52.62%   |
> | BioT5+                       | 54.06%   |
> | InstructProtein              | 80.40%   |
> | Mistral-7B-Instruct-v0.3     | 55.20%   |
> | Mistral-7B-Instruct-v0.3 ◇   | 76.00%   |
> | **△ Gain**               | **20.80%** |
> | Qwen3-14B                    | 54.20%   |
> | Qwen3-14B ◇                  | 81.20%   |
> | **△ Gain**               | **27.00%**  |
> | kimi-k2                      | 66.20%   |
> | kimi-k2 ◇                    | 85.40%   |
> | **△ Gain**               | **19.20%** |
> | GPT-4o                       | 55.00%   |
> | GPT-4o ◇                     | 78.00%   |
> | **△ Gain**               | **23.00%**  |
>
> >Q4. Domain expertise of human evaluators.
>
> We have added a clearer description in Appendix A. All evaluators have at least two years of research experience in bioinformatics or related biological fields.
>
> >Q5. Clarification of the term “zero-shot.”.
>
> We have revised the terminology throughout the manuscript, replacing “zero-shot” with the more precise wording (“without any parameter updates”).
>
> >Q6 Domain expertise of human evaluators.
>
> We thank the reviewer for pointing out the inconsistency. We have thoroughly checked all numerical results in the manuscript, and the mismatch in Figure 4 (left) has been corrected in the revised version.
>
> [1] Wang, Z., Zhang, Q., Ding, K., Qin, M., Zhuang, X., Li, X. and Chen, H., 2024, August. Instructprotein: Aligning human and protein language via knowledge instruction. In Proceedings of the 62nd Annual Meeting of the Association for Computational Linguistics (Volume 1: Long Papers) (pp. 1114-1136).

---

### Official Review · Reviewer_sf9o · 2025-11-02

**Soundness:** 3
**Presentation:** 2
**Contribution:** 3
**Rating:** 6
**Confidence:** 2

**Summary:**

Protein-as-Second-Language (PSL) framework is a method for protein function understanding using large language models (LLMs) without fine-tuning. The approach reformulates amino acid sequences as symbolic language and uses adaptive, bilingual context construction (sequence + natural language) to enable zero-shot reasoning. A curated dataset of ~80k protein–QA pairs spanning functional, descriptive, and reasoning tasks supports the method.

**Strengths:**

- Introduces the idea of treating protein sequences as a "second language" for LLMs, bridging symbolic biological and natural language reasoning.
- The bilingual corpus is diverse (79k QA pairs across 4 types), functionally rich, and biologically balanced.
- Works across frozen LLMs (3B–14B) and improves both open-source and proprietary models in zero-shot settings.

**Weaknesses:**

- No wet-lab or structure-level validation is presented; success is only measured by text-based QA metrics (e.g., ROUGE-L).

**Questions:**

- How does the method perform on out-of-distribution or rare protein families, especially those absent from the QA corpus?

---

> ### Author Response · Authors · 2025-12-03
>
> We thank the reviewer for the constructive assessment and for highlighting the key aspects that could be further clarified.
>
> >W1. Lack of structure-level validation; reliance on ROUGE-L
>
> To address this concern, we included a structure-level validation using AlphaFold 3. As detailed in the revised manuscript (Appendix, Figure 9), we compared GPT-4o’s structural descriptions with and without contextual examples across four representative proteins. The AF3-predicted structures provide residue-level confidence (pLDDT) and domain assignments, enabling an external check on whether the model captures biologically meaningful organization.
>
> Across all displayed cases, descriptions generated with our contextual examples aligned substantially better with AF3 predictions, including correct identification of catalytic regions, cofactor-binding pockets, and multi-domain arrangements. Zero-shot outputs, by contrast, often omitted these key structural elements. These results indicate that the benefits of our method extend beyond text-level matching and support improved biological coherence at the structural level.
>
>
> >Q1. Performance on low-similarity / out-of-distribution proteins
>
> We evaluate our method on proteins whose maximum sequence identity to the QA corpus is below 40%. This setting tests generalization to low-homology or out-of-distribution sequences, where close exemplars are unavailable.  As shown in the table below (the symbol ◇ denotes models augmented with our method), our method still yields a 7.12% ROUGE-L improvement, indicating that it remains effective even when no close homologs exist.
>
> | Model                     | ProtDescribe ROUGE-L | ProtDescribe BLEU-2 | ProtDescribe BERTScore | Protein2Text ROUGE-L | Protein2Text BLEU-2 | Protein2Text BERTScore | Mol-Instructions ROUGE-L | Mol-Instructions BLEU-2 | Mol-Instructions BERTScore |
> |---------------------------|------------------|-------------------|------------------|-------------------|-------------------|------------------|---------------|---------------|--------------|
> | Qwen2.5-3B                | 18.61            | 7.55              | 58.27            | 18.60             | 6.63              | 67.42            | 18.65         | 7.25          | 60.97        |
> | Qwen2.5-3B ◇              | 26.16            | 9.67              | 64.03            | 21.44             | 8.60              | 68.05            | 22.61         | 10.30         | 64.25        |
> | △ Gain                   | +7.55            | +2.12             | +5.76             | +2.84             | +1.97             | +0.63             | +3.96         | +3.05         | +3.28        |
> | Mistral-7B-Instruct-v0.3  | 17.04            | 6.84              | 60.08            | 16.28             | 5.89              | 65.08            | 11.44         | 3.83          | 55.44        |
> | Mistral-7B-Instruct-v0.3 ◇| 30.19            | 11.34             | 64.63            | 19.09             | 7.46              | 68.61            | 20.57         | 7.70          | 64.89        |
> | △ Gain                   | +13.15           | +4.50             | +4.55             | +2.81             | +1.58             | +3.53             | +9.13         | +3.87         | +9.45        |
> | Qwen3-14B                 | 23.72            | 10.52             | 63.24            | 17.84             | 6.23              | 68.31            | 13.53         | 3.56          | 53.18        |
> | Qwen3-14B ◇              | 36.12            | 11.09             | 65.51            | 22.51             | 10.19             | 70.28            | 15.73         | 5.90          | 60.19        |
> | △ Gain                   | +12.40           | +0.57             | +2.27             | +4.67             | +3.96             | +1.97             | +2.20         | +2.34         | +7.01        |
> | kimi-k2                   | 24.41            | 9.98              | 62.79            | 13.20             | 3.22              | 64.17            | 12.74         | 3.60          | 55.05        |
> | kimi-k2 ◇                | 35.68            | 10.56             | 65.99            | 17.09             | 5.15              | 67.59            | 18.77         | 5.53          | 65.30        |
> | △ Gain                   | +11.27           | +0.58             | +3.20             | +3.89             | +1.93             | +3.42             | +6.03         | +1.93         | +10.25       |
> | GPT-4o                    | 19.91            | 10.32             | 59.80            | 16.94             | 6.71              | 67.45            | 15.61         | 6.06          | 59.16        |
> | GPT-4o ◇                 | 34.08            | 11.00             | 63.40            | 23.38             | 10.02             | 70.93            | 21.70         | 8.14          | 65.32        |
> | △ Gain                   | +14.17           | +0.68             | +3.60             | +6.44             | +3.31             | +3.48             | +6.09         | +2.08         | +6.16        |

---

### Author Response · Authors · 2025-12-03
**Summary of Revisions**

We sincerely thank all reviewers for their constructive comments. We have revised the manuscript accordingly, with major updates summarized as follows (new or updated content highlighted in blue in the draft):

* We add results on low-similarity / OOD proteins (Table 2), as suggested by reviewers R1 & R2.
* We add more baselines, including some analogy/reasoning-augmented methods (Figure 7), as suggested by reviewer R4.
* We add computational overhead analysis (Table 3), as suggested by reviewers R3 & R4.
* We add structure-level validation with AF3 (Figure 9), as suggested by reviewer R1.
* We add formal equations detailing the context-exposure process (Section 3.2), as suggested by reviewers R3 & R4.

We hope our responses and revisions will address all reviewers' concerns!

---

### Meta-Review · Area_Chair_xNUb · 2026-01-07

**Summary:**

This paper proposes Protein-as-Second-Language, a training-free framework that enables frozen large language models to interpret protein sequences via retrieval-based in-context learning. By constructing bilingual contexts that pair amino-acid sequences with natural-language QA exemplars, the method reframes protein understanding as contextual analogy rather than supervised fine-tuning. The authors also curate a sizable bilingual corpus (~80k protein–QA pairs) to support this approach.

Strengths:
The reviewers agree that the idea of treating protein sequences as a “second language” and leveraging in-context learning without parameter updates is interesting and appealing. The bilingual protein–QA corpus is large, carefully curated, and shown to improve performance across multiple frozen LLMs.

Major Concerns:
1. Novelty and positioning: Some reviewers felt the novelty was incremental relative to prior in-context or retrieval-based protein modeling work (e.g., ProtEx), and that the initial submission under-emphasized related paradigms.
2. Evaluation rigor: Early concerns included over-reliance on ROUGE-L, lack of structural or biological validation, and possible data leakage due to Swiss-Prot overlap.
3. Generalization: Reviewers questioned robustness to low-homology or out-of-distribution proteins.
4. Clarity: Several reviewers requested more formal definitions of the retrieval, scoring, and context construction process.
5. Efficiency and scalability: Practical latency and comparison to fine-tuned or analogy-based methods were initially under-analyzed.

The authors have provided a rebuttal, addressing some of the concerns. However, some reviewers remain unconvinced about the degree of novelty and note that the work is primarily empirical rather than theoretical, as well as the method generalization. I believe this work requires another round of major revision before it can be published.

**Reviewer Concerns:**

see above

**Reviewer Scores:**

The submission received mixed but mostly borderline negative evaluations, with one clear reject, one weak accept, and two marginal reject reviews, yielding an average overall score below the acceptance threshold. The rebuttal has addressed some of the concerns, which could potentially improve their scores post discussion. However, the paper still needs a major revision before it can be accepted.

---

### Decision · Program_Chairs · 2026-01-26

Reject